# Aligning Visual Foundation Encoders to Tokenizers for Diffusion Models

**Bowei Chen**[1]  **Sai Bi**[2]  **Hao Tan**[2]  **He Zhang**[2]  **Tianyuan Zhang**[2]  **Zhengqi Li**[2]

**Yuanjun Xiong**[2]  **Jianming Zhang**[2]  **Kai Zhang**[2]

[1]University of Washington  [2]Adobe Research

https://aligntok.github.io

## ABSTRACT

In this work, we propose aligning pretrained visual encoders to serve as tokenizers for latent diffusion models in image generation. Unlike training a variational autoencoder (VAE) from scratch, which primarily emphasizes low-level details, our approach leverages the rich semantic structure of foundation encoders. We introduce a three-stage alignment strategy called *AlignTok*: (1) freeze the encoder and train an adapter and a decoder to establish a semantic latent space; (2) jointly optimize all components with an additional semantic preservation loss, enabling the encoder to capture perceptual details while retaining high-level semantics; and (3) refine the decoder for improved reconstruction quality. This alignment yields semantically rich image tokenizers that benefit diffusion models. On ImageNet 256×256, our tokenizer accelerates the convergence of diffusion models, reaching a gFID of 1.90 within just 64 epochs, and improves generation both with and without classifier-free guidance. Scaling to LAION, text-to-image models trained with our tokenizer consistently outperforms FLUX VAE and VA-VAE under the same training steps. Overall, our method is simple, scalable, and establishes a semantically grounded paradigm for continuous tokenizer design.

## 1 INTRODUCTION

Diffusion models have recently emerged as the leading method for high-fidelity image generation. A crucial component of training image diffusion models is the *continuous* visual tokenizer, which defines the latent space where diffusion operates (Rombach et al., 2022). Training such a tokenizer involves two tasks: (1) the encoder must learn a diffusion-friendly latent space, often referred to as the *diffusability* of the latent space (Skorokhodov et al., 2025); and (2) the decoder must learn to reconstruct the input signal. A common practice for training a continuous visual tokenizer is to adopt a variational autoencoder (VAE), optimized with reconstruction loss and a lightly weighted KL regularization term. Since the KL term typically has only a small weight, training is dominated by reconstruction loss, making the two tasks asymmetric: the decoder's reconstruction learning is direct and well-supervised, while the encoder's representation learning is indirect – the latent space is shaped largely as a byproduct of reconstruction and only weakly regularized by the KL prior. As a result, the latent space often develops an unpredictable structure dominated by low-level details, limiting its diffusability (Yao et al., 2025).

Recent work on tokenizers for image diffusion models has explored adding constraints such as semantic regularization (Fig. 1 left), which adds a loss term to encourage the latent space to align with representations from large pretrained encoders (Yao et al., 2025; Chen et al., 2025a). These studies demonstrate that semantically grounded latents exhibit better diffusability, leading to improved generation quality in diffusion models. However,

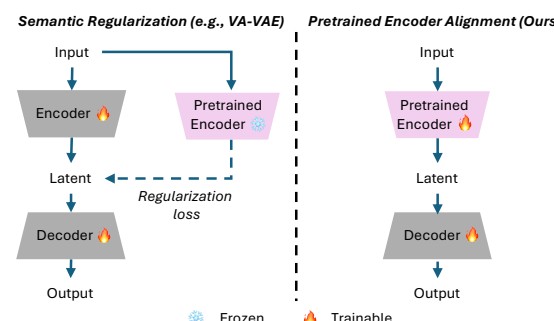

Figure 1: **Regularization vs. Alignment.**

these methods still fall short, as the encoder must still learn semantic structure *from scratch* via the regularization loss while simultaneously managing the competing reconstruction objective.

Our goal is to design an image tokenizer with stronger semantic grounding – hence better diffusability – with competitive reconstruction ability. Our intuition is that learning semantic is inherently more difficult than learning reconstruction. Thus, we take a different perspective: rather than forcing the encoder to learn semantics from scratch, we *align* a pretrained visual foundation encoder (*e.g.*, DINOv2 (Oquab et al., 2023)) to a visual tokenizer (Fig. 1 right). Since the encoder already captures rich semantic structure, our alignment makes the first task – learning a diffusion-friendly latent space (*i.e.*, achieving strong diffusability) – much easier. Training can then focus on equipping the tokenizer with reconstruction ability, avoiding the difficulties of learning semantic from scratch.

We implement this idea through a three-stage alignment procedure. First, we freeze the pretrained encoder and train a lightweight adapter and decoder with reconstruction loss, establishing a semantically grounded latent space. Second, we jointly optimize all components with an additional semantic preservation loss, enabling the encoder to capture fine-grained perceptual details essential for both reconstruction and generation, while maintaining its underlying semantic structure. Finally, we finetune only the decoder to enhance reconstruction fidelity while keeping the latent space fixed. This progressive alignment preserves a semantically rich, diffusion-friendly latent space that also retains the details necessary for accurate reconstruction and high-quality generation.

We demonstrate the effectiveness of our method on both the ImageNet 256×256 dataset (Deng et al., 2009) and the LAION dataset (Schuhmann et al., 2022) by training diffusion models using our continuous tokenizers. On ImageNet, our semantic tokenizer accelerates the convergence of diffusion models and improves generation quality over previous methods, both with and without classifier-free guidance (CFG), as well as in unconditional generation settings. To further validate scalability, we train text-to-image diffusion models on LAION and show that they converge significantly faster than the FLUX VAE (Labs, 2024) and VA-VAE (Yao et al., 2025). We conjecture that these benefits arise from a well-grounded semantic latent space, which provides a more structured representation.

In summary, we propose a new paradigm for training visual tokenizers by aligning pretrained visual encoders. Our approach is simple, scalable, and effective, and we believe it can open new directions in tokenizer design while inspiring future research in generative modeling.

## 2 RELATED WORK

### 2.1 FOUNDATION ENCODER FOR REPRESENTATION LEARNING

Large-scale foundation visual encoders enable the extraction of rich, transferable representations from diverse visual data. Models such as CLIP (Radford et al., 2021), SigLIP (Zhai et al., 2023), SigLIP 2 (Tschannen et al., 2025), MAE (He et al., 2022), Perception Encoder (Bolya et al., 2025), DINOv2 (Oquab et al., 2023), and DINOv3 (Siméoni et al., 2025) demonstrate that pretraining on massive datasets enables encoders to capture meaningful visual features that generalize effectively across downstream visual understanding tasks. In this paper, we adopt DINOv2 as our visual foundation encoder by default, since we empirically find that it is more effective for diffusion modeling than alternatives such as SigLIP 2 and MAE.

### 2.2 IMAGE TOKENIZERS FOR GENERATIVE MODELS

Image tokenizers are essential for scaling visual generation. They use an encoder–decoder framework to map images into compact latent spaces where generative models can operate more effectively and efficiently (Rombach et al., 2022). A common practice in training tokenizers is to rely on reconstruction losses such as L1 loss, perceptual loss, and adversarial loss (Rombach et al., 2022; Kouzelis et al., 2025; Wen et al., 2025; Wu et al., 2025; Yu et al., 2024; Skorokhodov et al., 2025; Hansen-Estruch et al., 2025; Silvestri & Ambrogioni, 2025; Van Den Oord et al., 2017). Depending on the design, tokenizers are either continuous, where the latent distribution is regularized by a KL loss, or discrete, where quantization with a codebook is enforced via a VQ loss. While these methods achieve faithful reconstruction fidelity, the resulting latent space is often dominated by fine-grained details because they mainly rely on reconstruction loss, which can hinder generative performance.

**Semantic Regularization.** Recent methods introduce semantic regularization strategies to enrich the image tokenizers with higher-level semantics (Xiong et al., 2025; Chen et al., 2025a; Yao et al.,

2025; Chen et al., 2025b; Lee et al., 2025; Yang et al., 2025; Chu et al., 2025; Lu et al., 2025). VA-VAE (Yao et al., 2025) introduces a continuous tokenizer for diffusion models by regularizing its latent space to be close to a pretrained encoder, whereas GigaTok (Xiong et al., 2025) proposes a discrete tokenizer for autoregressive models by imposing semantic constraints on decoder features. Instead of using semantic regularization, we align a pretrained encoder that is already capable of extracting high-level semantic representations, and show that this leads to a more diffusion-friendly latent space. We conduct extensive comparisons with VA-VAE, as both methods target diffusion models, and demonstrate that our alignment strategy outperforms semantic regularization.

**Pretrained Encoders as Discrete Tokenizers.** The use of pretrained encoders in tokenizers has also been studied, but primarily in the context of *discrete* tokenizers for *autoregressive models*. One line of work treats the pretrained encoder as a fixed feature extractor without fine-tuning it to encode perceptual details (Zheng et al., 2025a; Chen et al., 2025c; Xie et al., 2024; Wang et al., 2024). Another line of work fine-tunes pretrained encoders with additional architectural designs, such as introducing extra encoders (Qu et al., 2025; Jiao et al., 2025) or decoders (Han et al., 2025), or splitting encoder features into separate vocabularies (Song et al., 2025). A third direction leverages contrastive learning, fine-tuning the pretrained encoder with image–text supervision to capture semantic alignment (Wu et al., 2024; Ma et al., 2025; Zhao et al., 2025; Lin et al., 2025). However, this strategy primarily assumes that the encoder is a language-aligned model (*e.g.*, SigLIP 2).

In contrast to these works on *discrete* tokenizers, we focus on *continuous* tokenizers for diffusion models. Rather than introducing additional architectural design or requiring image–text supervision, we keep the simple architecture of autoencoder and directly align a pretrained encoder with a self-supervised semantic preservation loss. Our method can generalize to any visual encoders, offering a simple and scalable path toward semantically rich tokenizers for generative modeling.

**Concurrent Work**. (Tang et al., 2025) explores enabling pretrained CLIP with reconstruction ability. The key distinction is that we target diffusion-based generation, providing extensive experiments showing that aligning a pretrained encoder yields latent space with better diffusability than using semantic regularization. In contrast, their work focuses on unified multimodal understanding, generation, and editing within a hybrid architecture that combines multimodal large language models (MLLMs) with diffusion. Another difference is that we study different foundation visual encoders and identify DINOv2 as particularly well suited for latent diffusion models, whereas their focus remains on CLIP in the context of unified modeling.

**Relationship to RAE**. RAE Zheng et al. (2025b) proposes to directly adopt a pretrained foundation encoder, without fine-tuning, as the tokenizer encoder. While this preserves strong semantic representations, it results in a high-dimensional latent space with a large channel dimension. Diffusion models are known to struggle in such high-channel latent spaces, as optimization becomes unstable and noise scheduling becomes less effective. To mitigate this issue, RAE introduces techniques such as dimension-dependent noise shifting and a widened diffusion head to stabilize training in high-dimensional latents. Although RAE demonstrates strong performance for downstream generative model training, its reconstruction quality is limited by the frozen encoder, which is not optimized for faithful pixel-level reconstruction. In contrast, our approach fine-tunes the encoder, significantly improving reconstruction fidelity while maintaining semantic alignment. *We believe that combining RAE's high-channel semantic latent space with our alignment and fine-tuning strategy could offer the best of both worlds: strong semantic representations together with high-quality reconstruction.*

**Relationship to REPA-E**. While both REPA-E Leng et al. (2025) and our method leverage a vision foundation model, it is not straightforward to claim an overall advantage of one approach over the other because the two methods follow different setups and can in fact be combined. Specifically, REPA-E trains the tokenizer and the diffusion model jointly in an end-to-end manner. Even under this end-to-end regime, the authors show that initializing the tokenizer with an existing tokenizer works better than random initialization; specifically, their best model (E2E-VAE) initializes the tokenizer from VA-VAE. In contrast, our method focuses solely on training the tokenizer itself. This makes our tokenizer a drop-in replacement for the VA-VAE component inside REPA-E: since REPA-E already benefits from initializing its tokenizer with VA-VAE, it could similarly initialize with our tokenizer as a potentially stronger starting point for its end-to-end training.

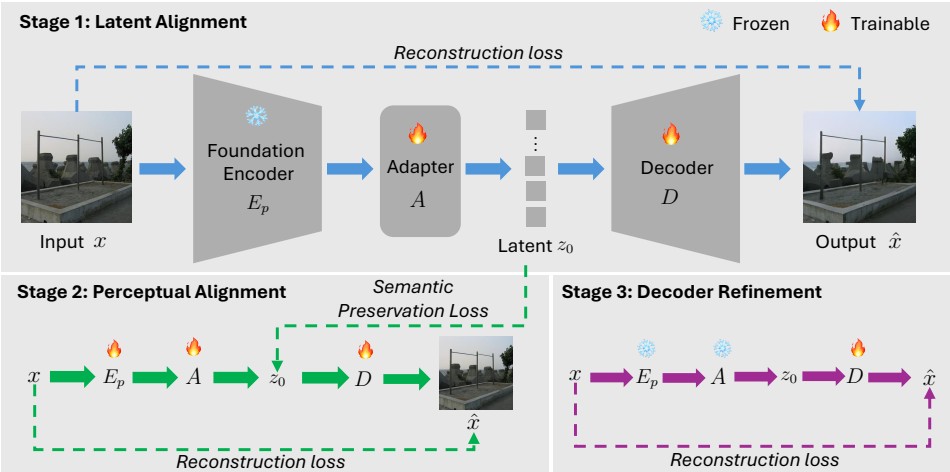

Figure 2: **Method Overview.** *Stage 1: Latent Alignment (top).* The pretrained encoder is kept frozen, while the adapter and decoder are trained with reconstruction loss to align its output into a semantically grounded latent space for generation. *Stage 2: Perceptual Alignment (bottom left).* All components are optimized jointly to enrich the latent space with low-level details, while a semantic preservation loss ensures that it retains high-level semantics. *Stage 3: Decoder Refinement (bottom right).* Only the decoder is fine-tuned with reconstruction loss to enhance reconstruction fidelity.

## 3  METHOD

We aim to build a semantic, diffusion-friendly visual tokenizer by aligning a pretrained visual encoder. We begin with a review of latent diffusion models, followed by an introduction of our method.

### 3.1  BACKGROUND OF LATENT DIFFUSION MODELS IN IMAGE GENERATION

Latent diffusion models (LDMs) (Rombach et al., 2022) operate by learning a denoising process in the compressed latent space produced by a continuous visual tokenizer. The tokenizer consists of an encoder $E$ and a decoder $D$. The encoder maps an input image $x \in \mathbb{R}^{H \times W \times 3}$ to a latent code $z_0 = E(x) \in \mathbb{R}^{\frac{H}{f} \times \frac{W}{f} \times d}$, where $H$ is image height, $W$ is image width, $f$ is the downsampling factor and $d$ is the channel dimension. The decoder reconstructs $\hat{x} = D(z_0)$. The tokenizer is trained with a reconstruction objective combining pixel-level L1, perceptual, and adversarial losses:

$$\mathcal{L}_{\text{rec}} = \mathcal{L}_{\ell_1}(x, \hat{x}) + w_p \, \mathcal{L}_{\text{perceptual}}(x, \hat{x}) + w_g \, \mathcal{L}_{\text{GAN}}(x, \hat{x}), \tag{1}$$

where $w_p$ and $w_g$ are the weights for the perceptual and adversarial loss, respectively. In addition, a KL regularization term is often included alongside the reconstruction loss to encourage a well-structured latent space. After training the tokenizer, a diffusion model learns to reverse a forward noising process in this learned latent space. A common formulation is flow matching, where we define an interpolating path:

$$z_t = (1 - t) \, z_0 + t \, z_1, \quad z_1 \sim \mathcal{N}(0, I), \ t \in [0, 1], \tag{2}$$

where $t$ is the diffusion timestep. The velocity field is given by $u_t = \frac{d}{dt} z_t = z_1 - z_0$. The diffusion model $v_\theta$ is trained to predict this velocity, with the loss $\mathcal{L}_{\text{FM}} = \mathbb{E}_{z_0, z_1, t} \left[ \| v_\theta(z_t, t) - u_t \|_2^2 \right]$.

### 3.2  ALIGNING PRETRAINED ENCODER TO VISUAL TOKENIZER

Our method leverages a pretrained visual encoder, which offers rich semantic representations, and progressively adapts it into a diffusion-friendly visual tokenizer. This is implemented through three alignment stages, as illustrated in Fig. 2. First, we align the encoder's semantic space to a generative latent space by training a lightweight adapter and decoder (*Latent Alignment*). Second, we jointly optimize all components to enhance generation and reconstruction fidelity while preserving semantic structure (*Perceptual Alignment*). Finally, we fine-tune the decoder to further improve reconstruction quality while keeping the latent space untouched (*Decoder Refinement*).

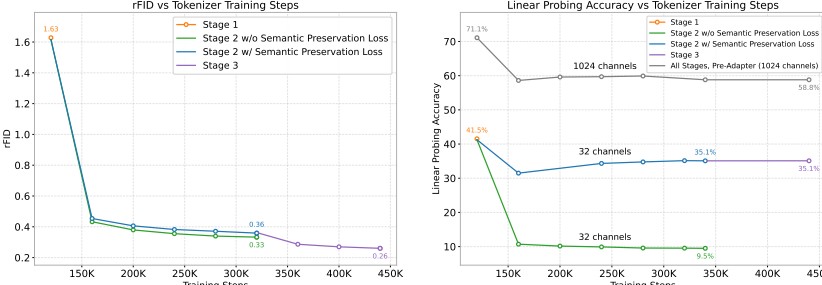

Figure 3: **Reconstruction vs. Semantic Preservation in Tokenizer Training.** *Left*: reconstruction FID (rFID) across training steps. *Right*: linear probing accuracy across training steps. Linear probing accuracy is evaluated on the latent code (32 channels), except for *All Stages, Pre-Adapter (1024 channels)*, which is reported only for reference. In this case, linear probing accuracy is measured on the feature before the adapter, using the same checkpoint as *Stage 2 w/ Semantic Preservation Loss*.

**Stage 1: Latent Alignment.** As the first stage, the goal is to adapt the pretrained encoder to create a latent space suitable for generation (Fig. 2 top). This requires that the latent representations contain semantic information and can be mapped back to the image domain for reconstruction.

Given an input image $x$, we extract its embedding using a frozen pretrained encoder $E_p$. Since embeddings from encoders trained for representation learning tasks are typically very high-dimensional (*e.g.*, 1024 channels for *DINOv2-L/14*), they are not directly suitable for diffusion models, which usually operate in lower dimensions (*e.g.*, 32 or 64). To address this, we introduce an adapter $A$ that projects the high-dimensional features into a compact latent code of dimension $d$ (32 by default):

$$z_0 = A(E_p(x)). \tag{3}$$

To complete the tokenizer, a decoder $D$ is then introduced to reconstruct the input image from $z_0$. During this stage, only the adapter $A$ and decoder $D$ are trained with the reconstruction loss in Eq. 1, while the pretrained encoder $E_p$ remains frozen to ensure a semantically-rich latent space. We omit the KL term because we found that it does not provide benefits and only imposes unnecessary distributional constraints, which can distort the encoder's semantics in the latent space.

Although this stage yields a semantically grounded latent space, it does not achieve high-fidelity reconstruction (see a noticeable color shift in top right image of Fig. 2 and the orange point in Fig. 3 left), as the frozen encoder cannot capture the fine-grained perceptual details required for precise image reconstruction and high-quality generation.

**Stage 2: Perceptual Alignment.** The goal of this stage is to adapt the pretrained encoder so that it can capture fine-grained, low-level image details while still preserving semantic features, as shown in Fig. 2 (bottom left). Starting from the checkpoints of the previous stage, we jointly optimize $E_p$, $A$, and $D$ using the same reconstruction loss as in Eq. 1. This encourages $E_p$ to encode richer details, thereby improving reconstruction quality, as indicated by the green curve in Fig. 3 (left). However, while reconstruction improves rapidly, this optimization simultaneously causes the latent space to catastrophically lose its semantic structure, as reflected by the sharp drop in linear probing accuracy (green curve in Fig. 3 right). To address this issue, we introduce a simple yet effective semantic preservation loss, which constrains the latent codes produced in the current stage to remain close to those obtained in the previous stage. Formally, we define this loss as an L2 loss:

$$\mathcal{L}_{\text{sp}} = L_{\ell_2}(z_0^*, z_0), \tag{4}$$

where $z_0^*$ and $z_0$ are the latent codes produced by $E_p$ and $A$ in the current stage (being updated) and the previous stage (kept frozen), respectively. An alternative is to apply this loss directly on the output of $E_p$, providing more flexibility by leaving the adapter $A$ unregularized. However, we found this variant degrades generation quality (see ablation study). The overall loss for this stage is:

$$\mathcal{L}_{\text{pa}} = \mathcal{L}_{\text{rec}} + w_{sp}\mathcal{L}_{\text{sp}}, \tag{5}$$

where $w_{sp} = 1$ balances the semantic preservation loss. As shown in Fig. 3 (blue curve), this strategy maintains a semantically rich latent space while achieving comparable reconstruction performance.

**Stage 3: Decoder Refinement.** The previous two stages already align the pre-trained encoder into a visual tokenizer that significantly boosts generation performance. The goal of this stage is to further

Table 1: **Ablation study.** Evaluated on ImageNet 256×256 at 80K training steps with 30 sampling steps, using the CFG scale that yields the lowest generation FID (gFID). Our full three-stage model achieves the best balance between reconstruction and generation quality.

| Configuration | rFID↓ | PSNR↑ | LPIPS↓ | L. P. Acc.↑ | gFID↓ | IS↑ | Prec↑ | Recall↑ |
|---|---|---|---|---|---|---|---|---|
| *Semantic Preservation Loss* | | | | | | | | |
| Weight = 0 | 0.33 | **26.29** | **0.110** | 9.50% | 3.05 | 215.1 | 0.832 | 0.550 |
| Weight = 5 | 0.49 | 23.70 | 0.163 | 40.55% | 2.48 | 244.4 | 0.836 | 0.566 |
| Weight = 10 | 0.59 | 22.63 | 0.189 | 40.89% | 2.59 | 243.5 | 0.839 | 0.562 |
| Applied Pre-Adapter | 0.34 | 25.78 | 0.118 | 15.61% | 2.83 | 226.9 | 0.835 | 0.553 |
| Cosine Loss | 0.37 | 25.23 | 0.129 | 37.99% | 2.23 | 248.6 | 0.819 | 0.585 |
| *Training Strategy for Stage 2* | | | | | | | | |
| LoRA Fine-Tuning | 1.35 | 26.18 | 0.121 | 18.56% | 2.97 | 243.3 | 0.815 | 0.557 |
| w/o EMA | 0.33 | 25.70 | 0.122 | 35.04% | 2.24 | 246.0 | 0.809 | 0.587 |
| *High-Level Design* | | | | | | | | |
| Larger Decoder | 0.36 | 26.27 | 0.121 | 27.24% | 2.52 | 248.3 | 0.808 | 0.571 |
| Stage 1 only | 1.63 | 17.34 | 0.323 | **41.53%** | 3.00 | 246.4 | **0.843** | 0.529 |
| Stage 1 + Stage 2 | 0.36 | 25.62 | 0.121 | 35.09% | 2.19 | 248.6 | 0.811 | 0.591 |
| Full Model | **0.26** | 25.83 | 0.117 | 35.09% | **2.17** | **249.3** | 0.811 | **0.599** |

refine the decoder to improve reconstruction quality, as shown in Fig. 2 (bottom right). The key motivation is that, although the latent space is semantically aligned, the decoder may still underfit because the latent space kept changing throughout the previous two stages. Fine-tuning the decoder alone allows it to better exploit the existing latent representation without disturbing its semantic structure. Specifically, we continue training from the second stage but only update the decoder. This preserves the learned latent space and can even be applied after training the downstream generative model. As shown in Fig. 3 left (purple curve), this stage improves reconstruction performance.

# 4 EXPERIMENTS

We evaluate on two datasets: ImageNet 256×256 (Deng et al., 2009) and a subset of LAION-2B (Schuhmann et al., 2022). Most ablation studies and baseline comparisons are conducted on ImageNet (Sec. 4.1–4.3), followed by larger scale text-to-image experiments on LAION (Sec. 4.4). For ImageNet, we set the downsampling ratio to $f = 16$, the latent channel dimension to $d = 32$, the semantic preservation loss weight to $w_{sp} = 1$, and the sampling step to 30, unless otherwise specified. Same as VA-VAE (Yao et al., 2025), we use LightningDiT ($\sim$673M parameters) as generative model for ImageNet experiments. See more implementation details in Appendix (Sec. A).

**Metrics.** For ImageNet, we use reconstruction FID (rFID), PSNR, and LPIPS to evaluate reconstruction quality; linear probing accuracy to assess the semantic structure of latent space; and generation FID (gFID), Inception Score (IS), Precision (Prec), and Recall to measure generative performance.

## 4.1 ABLATION STUDY

We test different variants of our design and summarize them in Tab. 1. *All variants we test do not employ the third stage, thus we mainly compare them to Stage 1 + Stage 2 for analysis.*

**Semantic Preservation Loss.** We first analyze the effect of varying the weight of the semantic preservation loss in Eq. 5. Without this loss (weight = 0), the model achieves slightly better reconstruction quality, but the linear probing accuracy and generative metrics degrade, indicating that the latent space collapses toward low-level details at the expense of semantic structure. Increasing the weight (5 or 10) enforces stronger preservation, which improves generative performance over the one with weight = 0, but comes with a notable drop in reconstruction fidelity. These results reveal a clear trade-off: too little preservation leads to semantic drift, whereas too much preservation overly constrains the encoder and harms pixel-level fidelity. In the *Stage 1 + Stage 2* variant, a moderate weight of 1 achieves the best balance between generation and reconstruction.

We test a variant *Applied Pre-Adapter* where the semantic preservation loss is applied directly on the outputs of the pretrained encoder (before adapter). This approach partially preserves semantics, and the generation quality lags behind *Stage 1 + Stage 2* variant. This may be because leaving the adapter unregularized gives it too much flexibility, leading to a loss of semantic structure. In addition, we also test a variant *Cosine Loss* that replaces the L2 loss with a cosine similarity loss, which applies the semantic constraint with explicit normalization. This variant achieves performance comparable

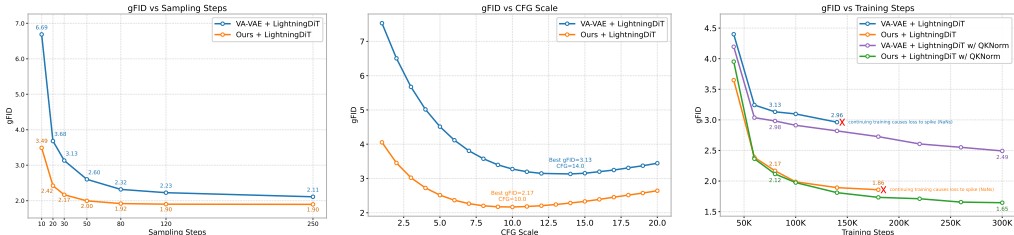

Figure 4: **Comparison of Sampling Steps, CFG Scales, and Convergence Speed.** Evaluated on ImageNet 256×256. *Left*: effect of sampling steps versus gFID at 80K training steps. *Middle*: effect of CFG scale versus gFID at 80K training steps with 30 sampling steps. *Right*: effect of training steps versus gFID with 30 sampling steps. QKNorm is enabled during extended training to ensure stability. All gFIDs in the left and right figures are reported using the best-searched CFG scale.

to L2 loss. This suggests that, at least in our current setup, both losses lead to similar behavior, indicating that the performance differences are not dominated by the specific choice of L2 versus cosine loss.

**Training Strategy.** We compare different optimization strategies for the perceptual alignment stage against the *Stage 1 + Stage 2* variant. *LoRA Fine-Tuning* performs noticeably worse, showing that restricting updates to low-rank adapters is insufficient for balancing semantics and reconstruction. Removing EMA (Karras et al., 2024) slightly decreases generation performance. The *Stage 1 + Stage 2* variant uses EMA to stabilize training and thereby produce a more stable latent space.

**High-Level Design.** We also evaluate alternative architectural and training-stage designs. Employing a larger decoder (∼351M parameters), implemented as a hybrid transformer–CNN architecture following (Xiong et al., 2025), yields slight improvements in reconstruction while degrading generative performance. This suggests that increasing decoder capacity may not yield additional benefits when training on a dataset of ImageNet's scale. Using only the latent alignment stage (stage 1) preserves high-level semantics but causes poor reconstruction, rendering this configuration unsuitable for generative modeling. Introducing the perceptual alignment stage (stage 2) resolves this issue, substantially improving both reconstruction and generation, thereby validating the need to fine-tune the encoder. Extending to the three-stage design with decoder refinement further strengthens reconstruction and slightly improves generative quality, showing the importance of progressive alignment.

**Other Pretrained Encoders.** We compare different pretrained encoders as the backbone of our tokenizer. As shown in Tab. 2, *MAE* achieves the strongest reconstruction fidelity but performs the worst in generation, likely due to its reconstruction objective. *DINOv2* achieves the best balance, delivering superior generation quality while maintaining competitive reconstruction performance.

Table 2: **Comparison of Various Pretrained Encoders.** ImageNet 256×256 at 80K steps; 30 sampling steps; CFG tuned for best gFID. Stage 3 is not applied.

| Config. | rFID↓ | gFID↓ | IS↑ | Prec↑ | Recall↑ |
|---|---|---|---|---|---|
| MAE | **0.29** | 3.12 | 216.5 | **0.834** | 0.543 |
| SigLIP 2 | 0.35 | 2.22 | 246.1 | 0.816 | 0.576 |
| DINOv2 | 0.36 | **2.19** | **248.6** | 0.811 | **0.591** |

## 4.2 Comparison with Other Tokenizers

We provide a comprehensive comparison with two baselines: the vanilla VAE and VA-VAE (Yao et al., 2025), which is a representative method that applies semantic regularization using a pretrained DINOv2 model to the latent space. Unless otherwise noted, all baseline checkpoints are taken from the official VA-VAE implementation.

### 4.2.1 Class-Conditional Generation

**Sampling Step.** In Fig. 4 left, we show that our method achieves better performance than VA-VAE with fewer sampling steps. While VA-VAE requires more than 120 steps to approach its best FID, our tokenizer reaches near-optimal performance with 80 steps. Remarkably, our 50-step generations even surpass the quality of VA-VAE's outputs at 250 steps – the default in its official implementation. We attribute this to the smoother latent space, where discretization errors cause only minor variations, rather than drastic shifts in color, semantics, or overall composition. This robustness allows the model to maintain higher fidelity with fewer sampling steps.

Table 3: **Comparison with Other Tokenizers.** Evaluated on ImageNet 256×256 at 80K training steps with 30 sampling steps. The checkpoints for both the Vanilla VAE and VA-VAE with CNN encoders are taken from the official VA-VAE repository. *VA-VAE*[†] denotes the VA-VAE model we trained, using a ViT encoder that matches our architecture but initialized from scratch.

| Tokenizer | Enc. Arch. | rFID↓ | L. P. Acc.↑ | gFID (uncond)↓ | gFID w/o CFG↓ | gFID w/ CFG↓ |
|---|---|---|---|---|---|---|
| *f16d32 (downsampling factor 16, latent dimension 32)* | | | | | | |
| Vanilla VAE | CNN | **0.26** | 6.04% | 29.12 | 10.17 | 3.31 |
| VA-VAE | CNN | 0.28 | 22.96% | 19.12 | 7.79 | 3.13 |
| VA-VAE[†] | ViT | 0.37 | 33.57% | 18.27 | 8.21 | 3.16 |
| Ours | ViT | **0.26** | **35.09%** | **11.80** | **4.05** | **2.17** |
| *f16d64 (downsampling factor 16, latent dimension 64)* | | | | | | |
| Vanilla VAE | CNN | 0.17 | 5.09% | 36.41 | 16.99 | 4.03 |
| VA-VAE | CNN | **0.14** | 19.72% | 26.70 | 12.23 | 3.20 |
| VA-VAE[†] | ViT | 0.18 | 43.53% | 19.81 | 7.92 | 3.19 |
| Ours | ViT | 0.17 | **46.99%** | **14.40** | **5.24** | **2.34** |

**CFG Scale.** In Fig. 4 middle, we plot gFID versus CFG scale, where our tokenizer consistently outperforms VA-VAE across the entire range. Notably, our method achieves strong performance even at low CFG values, whereas VA-VAE relies on larger guidance scales to reach comparable fidelity. This shows that our latent space already encodes well-separated class semantics, reducing the dependence on aggressive guidance and yielding better generations with smaller CFG.

**Convergence Speed.** We compare the convergence speed of generative model training using our tokenizer against VA-VAE. As shown in Fig. 4 right, our approach consistently achieves better gFID across training iterations. By aligning with a pretrained encoder rather than relying on regularization, our method provides a more semantically structured latent space. This leads to roughly 5x faster training, requiring only ∼60K steps compared to VA-VAE's 300K steps for comparable quality, when evaluated with 30 sampling steps. See Appendix for qualitative comparisons (Sec. C.3).

**Different Channel Dimensions.** In Tab. 3, we compare Vanilla VAE, VA-VAE, and our method using latent spaces with 32 and 64 channels. Key observations include: (1) Vanilla VAE performs worst in terms of gFID. This is because its latent space primarily encodes low-level details, which diffusion models struggle to exploit effectively. (2) Our method consistently outperforms VA-VAE in terms of gFID, regardless of whether CFG is used. (3) When VA-VAE employs the same ViT encoder and is trained for the same number of steps as ours, it achieves comparable linear probing accuracy but still falls short in generative performance. This suggests that our semantic structure provides advantages beyond class separation, yielding a more semantically organized latent space that boosts the generative performance of diffusion models.

### 4.2.2 UNCONDITIONAL GENERATION

We also evaluate our method in the unconditional setting (class number = 1), as shown in Tab. 3. Our semantic latent space consistently outperforms baseline tokenizers, producing higher-quality generations without relying on class information. It is worth noting that all models are trained for only 80K steps, so the results may not reflect fully optimized performance.

### 4.2.3 RECONSTRUCTION

As shown in Tab. 3, our method achieves competitive reconstruction with 32 channels but lags behind VA-VAE (CNN encoder) at 64 channels. Lowering the semantic preservation loss weight and increasing learning rate in stage 2 improves reconstruction to match VA-VAE, with only a slight drop in generation performance (still surpassing VA-VAE). See Appendix (Sec. C.2) for details.

### 4.3 COMPARISON WITH OTHER SYSTEMS

We conduct a systematic comparison with other systems, as shown in Tab. 4. Both VA-VAE and our method are sampled using 250 sampling steps. When comparing our method to VA-VAE at 64 epochs (80K training steps), we surpass it in both reconstruction and generation quality, highlighting our superior convergence speed. For the 800-epoch setting (1M training steps), we retrain

Table 4: **System-Level Comparison.** We compare with VAR (Tian et al., 2024), MagViT-v2 (Yu et al., 2023), MAR (Li et al., 2024), l-DeTok (Yang et al., 2025), MaskDiT (Zheng et al., 2023), DiT (Peebles & Xie, 2023), SiT (Ma et al., 2024), FasterDiT (Yao et al., 2024), MDT (Gao et al., 2023a), MDTv2 (Gao et al., 2023b), REPA (Yu et al., 2025), CausalFusion (Deng et al., 2024), MAETok (Chen et al., 2025a), and VA-VAE (Yao et al., 2025). Gray and purple regions refer to LightningDiT trained for 64 epochs (80K training steps, no QKNorm) and 800 (1M training steps, with QKNorm) epochs, respectively. Bold numbers indicate the best result in each color block.

| Method | Tokenizer | # token × # dim | rFID↓ | Training Epochs | Gen w/o CFG | | | | Gen w/ CFG | | | |
|---|---|---|---|---|---|---|---|---|---|---|---|---|
| | | | | | gFID ↓ | IS ↑ | Prec ↑ | Recall ↑ | gFID ↓ | IS ↑ | Prec ↑ | Recall ↑ |
| *AutoRegressive (AR)* | | | | | | | | | | | | |
| VAR-d30 | - | 256 × 32 | - | 350 | - | - | - | - | 1.92 | 323.1 | 0.82 | 0.59 |
| MagViT-v2 | - | 256 × 5 | - | 1080 | 3.65 | 200.5 | - | - | 1.78 | 319.4 | - | - |
| MAR | LDM | 256 × 16 | 0.53 | 800 | 2.35 | 227.8 | 0.79 | 0.62 | 1.55 | 303.7 | 0.81 | 0.62 |
| MAR-L | l-DeTok | 256 × 16 | 0.68 | 800 | 1.86 | 238.6 | 0.82 | 0.61 | 1.35 | 304.1 | 0.81 | 0.62 |
| *Diffusion Transformer Using SD-VAE* | | | | | | | | | | | | |
| MaskDiT | SD-VAE | 1024 × 4 | 0.61 | 1600 | 5.69 | 177.9 | 0.74 | 0.60 | 2.28 | 276.6 | 0.80 | 0.61 |
| DiT | | | | 1400 | 9.62 | 121.5 | 0.67 | 0.67 | 2.27 | 278.2 | 0.83 | 0.57 |
| SiT | | | | 1400 | 8.61 | 131.7 | 0.68 | 0.67 | 2.06 | 270.3 | 0.82 | 0.59 |
| FasterDiT | | | | 400 | 7.91 | 131.3 | 0.67 | 0.69 | 2.03 | 264.0 | 0.81 | 0.60 |
| MDT | | | | 1300 | 6.23 | 143.0 | 0.71 | 0.65 | 1.79 | 283.0 | 0.81 | 0.61 |
| MDTv2 | | | | 1080 | - | - | - | - | 1.58 | 314.7 | 0.79 | 0.65 |
| REPA | | | | 800 | 5.90 | - | - | - | 1.42 | 305.7 | 0.80 | 0.65 |
| CausalFusion | | | | 800 | 3.61 | 180.9 | 0.75 | 0.66 | 1.77 | 282.3 | 0.82 | 0.61 |
| *LightningDiT without QKNorm* | | | | | | | | | | | | |
| LightningDiT | MAETok | 128 × 32 | 0.48 | 320 | 2.21 | 208.3 | - | - | 1.73 | 308.4 | - | - |
| LightningDiT | VA-VAE | 256 × 32 | 0.28 | 800 | 2.17 | 205.6 | 0.77 | 0.65 | 1.35 | 295.3 | 0.79 | 0.65 |
| *LightningDiT with QKNorm* | | | | | | | | | | | | |
| LightningDiT | VA-VAE | 256 × 32 | 0.28 | 800 | 2.50 | 206.2 | **0.76** | 0.65 | 1.52 | 286.6 | **0.79** | **0.65** |
| LightningDiT | AlignTok | 256 × 32 | **0.26** | 800 | **2.04** | 206.2 | **0.76** | **0.67** | **1.37** | **293.6** | **0.79** | **0.65** |
| *LightningDiT without QKNorm* | | | | | | | | | | | | |
| LightningDiT | VA-VAE | 256 × 32 | 0.28 | 64 | 5.14 | 130.2 | 0.76 | **0.62** | 2.11 | 252.3 | 0.81 | 0.58 |
| LightningDiT | AlignTok | 256 × 32 | **0.26** | 64 | **3.71** | **148.9** | **0.77** | **0.62** | **1.90** | **260.9** | **0.81** | **0.61** |

Table 5: **Quantitative Comparison on Text-to-Image (T2I) Generation with FLUX VAE.** Compared on *COCO Prompt 6K*, which has 6K captions sampled from the COCO validation set. Each 2B-parameter T2I model is trained for 100K steps and evaluated at 256×256 resolution with CFG. rFID is computed using 200K randomly sampled images from the COYO-700M dataset (Minwoo et al., 2022).

| Tokenizer | rFID | gFID | KID | HPSv2 | PickScore | ImageReward | Aesthetic Scores | CLIP Scores | VQA Scores |
|---|---|---|---|---|---|---|---|---|---|
| FLUX VAE | **0.102** | 35.78 | 0.018 | 0.242 | 0.397 | 0.162 | 5.411 | 31.21 | 0.775 |
| AlignTok | 0.443 | **30.27** | **0.015** | **0.249** | **0.603** | **0.564** | **5.573** | **32.21** | **0.849** |

LightningDiT of VA-VAE using the official repository with QKNorm enabled – necessary to avoid loss spikes, but slightly degrade generative performance, as noted by the authors. Our method (with QKNorm) achieves a gFID of 1.37, outperforming VA-VAE's 1.52 (with QKNorm) and comparable to the original VA-VAE result of 1.35 (without QKNorm) reported in their paper.

## 4.4 SCALE-UP EXPERIMENTS ON TEXT-TO-IMAGE GENERATION

We conduct a system-level comparison of our tokenizer with the widely adopted FLUX VAE on the text-to-image generation task. We train our tokenizer on images resized so that their shortest edge is 256 pixels, preserving aspect ratios. The diffusion model is trained for 200K steps at 256 resolution and fine-tuned for 90K steps at 512 resolution, both with variable aspect ratios.

For comparison, Fig. 5 presents results from generative models trained with our tokenizer and with FLUX VAE for 100K steps at 256×256 resolution. Our method produces images with better coherence, stronger text alignment, and competitive visual quality. Quantitative results (Tab. 5) further confirm the advantage of our approach. Fig. 6 presents our generated results at 512 resolution. Notably, our tokenizer is trained only on 256-resolution images, demonstrating its ability to generalize to unseen resolutions. These results suggest that our approach scales effectively and can potentially serve as a strong alternative to FLUX VAE in large-scale training. See Appendix (Sec. C.3-C.5) for additional quantitative results on two more prompt sets, as well as qualitative results, including convergence speed comparisons and generated images across different resolutions and aspect ratios.

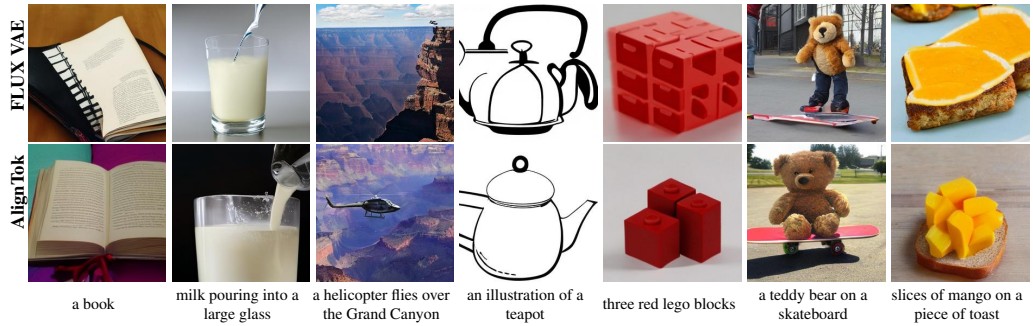

Figure 5: **Qualitative Comparison on Text-to-Image Generation with FLUX VAE.** Input text prompts are shown below the images and results (256×256 resolution) are generated from generative models trained for 100K steps. Our method (bottom row) produces images with better coherence and prompt alignment compared to the one using FLUX VAE (top row).

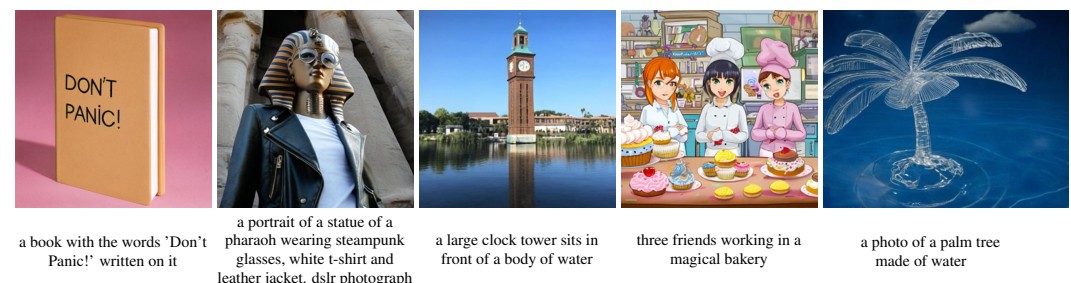

Figure 6: **Qualitative Results of Our Method on Text-to-Image Generation at 512 Resolution.** The input text prompts are shown below the images. Results are obtained from diffusion models trained for 290K steps. The first four are square images, and the final one has a 4:5 aspect ratio.

## 5 LIMITATIONS AND DISCUSSIONS

Our method, while effective, has several limitations. First, although the semantic latent space improves generative quality, its reconstruction ability still lags behind FLUX VAE. This gap could be narrowed with stronger decoders, larger channel dimensions, longer training, or scaling to larger models and compute budgets. Second, our evaluation is limited to images up to 512 resolution. Exploring higher resolutions is an interesting future direction, and the recently introduced DINOv3 – showing strong capability for variable resolutions – could be leveraged for this purpose.

Our study highlights a key insight: aligning a pretrained semantic encoder yields a more generation-friendly latent space than learning semantic from scratch. Although our work focuses on tokenizers for image diffusion, extending this approach to video tokenization, discrete tokenizers for autoregressive generation, and unified representations for multi-modal models is a promising direction for future work. We hope our findings inspire a rethinking of tokenizer design in generative modeling.

**Acknowledgements.** This work was done while Bowei Chen and Tianyuan Zhang were interning at Adobe. We would like to thank Hailin Jin and Jingwan (Cynthia) Lu for their valuable support throughout the internship.

**Reproducibility Statement.** We provide the implementation details and hyperparameters in Sec. A of the Appendix, which are sufficient to reproduce the results of our method.

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

# A    IMPLEMENTATION DETAILS

**Tokenizer.** For ImageNet, we set the downsampling ratio to $f = 16$, the latent channel dimension to $d = 32$. We use the pretrained DINOv2 checkpoint *vit_large_patch14_dinov2.lvd142m* ($\sim$304M parameters) (Oquab et al., 2023) as encoder. The adapter is implemented as a two-layer MLP that projects the 1024-dimensional encoder output into a 32-dimensional latent space. The decoder is a convolutional network with $\sim$42M parameters, same as VA-VAE (Yao et al., 2025). The training images are first resized so that the shorter edge is 256 pixels while preserving the aspect ratio, and then randomly cropped to 256$\times$256. Since the DINOv2 encoder uses a patch size of 14, we first resize the input image to 224$\times$224 resolution before feeding it into the encoder. This produces a 16$\times$16 latent feature map, matching the downsampling ratio of $f = 16$ for 256-resolution images. For training, we use a batch size of 64 and train on 8 NVIDIA H100 GPUs. The learning rate is set to 1e-4 for the first and third stages, and 1e-5 for the second stage. Following (Yao et al., 2025), the generator loss in $L_{\text{GAN}}$ is rescaled to match the magnitude of $L_{\ell_1} + w_p L_{\text{perceptual}}$ based on the ratio of their gradient norms at the last layer of the decoder, where $w_p = 1.0$. Similarly, we rescale $L_{sp}$ according to the gradient ratio at the last layer of the encoder. We set $w_g = 0.5$ and $w_{sp} = 1$, which are applied after loss rescaling. We enable $L_{\text{GAN}}$ after 5K training steps in the first stage, and keep it active throughout the second and third stages. We enable Exponential Moving Average (EMA) (Karras et al., 2024) during training and apply them in the second and third stages during inference. Training runs for 6 epochs in the first stage ($\sim$120K steps, $\sim$7 hours), 11 epochs in the second stage ($\sim$220K steps, $\sim$24 hours), and 5 epochs in the third stage ($\sim$100K steps, $\sim$6 hours).

For LAION dataset, we use the pretrained DINOv2 checkpoint *vit_large_patch14_reg4_dinov2* ($\sim$304M parameters) (Schuhmann et al., 2022) as encoder. The tokenizer is trained with a batch size of 256 and a learning rate of 1e-4 for all stages, using 32 NVIDIA H100 GPUs. We set $w_{sp}$ to 3. All other settings follow those of the ImageNet experiments. Training images are resized such that the shortest edge is 256 pixels while preserving the aspect ratio, without restricting them to square shapes. Similar to before, the input images are resized to multiples of 14 so that the encoder output aligns with the downsampling ratio of $f = 16$. We train for 300K steps in the first stage ($\sim$30 hours), 100K steps in the second stage ($\sim$20 hours), and 50K steps in the last stage ($\sim$5 hours).

**Generative Models.** For ImageNet experiments, we use LightningDiT (Yao et al., 2025) as the generative model ($\sim$673M parameters) and adopt the same hyperparameters as in its official implementation. We set the batch size to 1024, the learning rate to 2e-4, and the transformer patch size to 1. Unless otherwise specified, all experiments are trained for 80K steps (64 epochs), which takes approximately 12 hours on 8 NVIDIA H100 GPUs. We do not enable QKNorm for models trained with fewer than 200K steps, following the official implementation. When training models for more than 200K steps, we enable QKNorm from the start of the training to stabilize optimization and prevent loss spikes. We use the Euler sampler with 30 sampling steps unless otherwise specified. Following LightningDiT, we apply CFG to only the first three latent channels for a fair and consistent comparison, unless otherwise specified.

For LAION experiments, we employ a diffusion transformer (2B parameters) following the FLUX implementation (Labs, 2024; Zhang et al., 2024; Kai et al., 2025a;b). We use a learning rate of 1e-4 with weight decay and a transformer patch size of 1. Training begins on 256-resolution images with varying aspect ratios for 200K steps using a batch size of 2048, which requires roughly 140 hours on 32 H100 GPUs. We then continue on 512-resolution images with varying aspect ratios for an additional 90K steps using a batch size of 512, requiring about 135 hours on 32 H100 GPUs. For inference, we apply EMA checkpoints and use a DDIM sampler (Song et al., 2020) with 50 steps, setting the classifier-free guidance (CFG) scale to 5.

**Datasets.** For our text-to-image training dataset LAION-2B, we apply a series of pre-filters to remove low-quality or undesired samples: we exclude images that are NSFW, watermarked, low-aesthetic, invalid, or from blocked domains, resulting in a subset of about 616M images.

**Linear Probing.** We follow the standard ImageNet protocol widely used to assess the semantic quality of latent representations. Specifically, we freeze both the VAE encoder and the adapter, and train a single linear classifier on top of the adapter's output features to predict ImageNet-1K classes. We use linear probing because it directly measures how much semantic information is linearly accessible in the latent space, making it a widely adopted proxy for semantic quality. The

classifier is trained for 3 epochs using Adam optimizer with a learning rate of 0.001 and a batch size of 256 on the ImageNet-1K training set, and evaluated on the ImageNet validation set.

**Ablation Study.** All methods, except for the *Full Model*, are trained without the third stage (*i.e.*, decoder refinement). For the *Larger Decoder* variant, we replace the CNN decoder with a larger decoder composed of a ViT architecture followed by a CNN module (totaling ∼351M parameters), adapted from (Xiong et al., 2025). For the variant *LoRA Fine-Tuning*, we apply LoRA to both the attention and MLP layers, using a rank of 16, an alpha of 32, and a dropout of 0.1. All variants (except for *LoRA Fine-Tuning*) are trained with the same hyperparameters and training iterations for fair comparison: 6 epochs in the first stage and 11 epochs in the second stage. The *Full Model* is additionally trained with a third stage of 6 epochs. These settings follow the default configuration described in the main paper.

**Comparison with Other Pretrained Encoders.** All methods are evaluated without training Stage 3. We use "vit_large_patch16_224.mae" and "google/siglip2-large-patch16-256" as the pretrained encoders for the *MAE* and *SigLIP 2* variants, respectively. For the *SigLIP 2* variant, we set $w_{sp} = 2$, as this yields better results. Apart from this adjustment, all methods are trained with identical hyperparameters and iterations, following the default configuration outlined in the main paper.

**Comparison with Other Tokenizers.** Our method with 32 channels is trained following the default configuration described in the main paper. For the 64-channel version, we train for 6 epochs (∼120K steps) in the first stage, 17 epochs (∼340K steps) in the second stage, and 11 epochs (∼220K steps) in the third stage. For the *Vanilla VAE* and *VA-VAE* with a CNN encoder (∼28M parameters), we use the pretrained checkpoints from (Yao et al., 2025) for both the 32- and 64-channel experiments. All pretrained checkpoints were trained with a batch size of 256 for 50 epochs, except the 32-channel VA-VAE model, which was trained for 125 epochs. For *VA-VAE* with a ViT encoder, we adopt the same architecture as our method but without initializing it from DINOv2 weights, and use the learning rate of 1e-4. The 32-channel version is trained for 22 epochs, whereas the 64-channel version is trained for 34 epochs, matching the number of epochs used in our model training.

**Scale-Up Experiments on Text-to-Image Generation.** To compute rFID, we use 200K images randomly sampled from the COYO-700M (Minwoo et al., 2022) dataset. The gFID and KID (Bińkowski et al., 2018) on *COCO Prompt 6K* are computed using 6K generated images and 6K corresponding real images from the sampled captions.

For completeness, we provide the reference for the metrics we used for evaluation: FID (Heusel et al., 2017), KID (Bińkowski et al., 2018), HPSv2 (Wu et al., 2023), PickScore (Kirstain et al., 2023), ImageReward (Xu et al., 2023), Aesthetic Scores (Schuhmann et al., 2022), CLIP Scores (Hessel et al., 2021), VQA Scores (Lin et al., 2024), LPIPS (Zhang et al., 2018).

# B  TRAINING AND INFERENCE COST

We show cost comparison of training in Tab. 6. Our main observations are as follows:

(1) **Total GPU memory**. The vanilla VAE consumes the least memory because it does not load any pretrained encoder. Our Stage 1 and Stage 3 use less memory than VA-VAE since our encoder is frozen during these stages, whereas VA-VAE loads CNN encoder (trainable) + DINOv2 encoder (frozen). Our Stage 2 consumes the most memory because it fine-tunes the DINOv2 encoder.

(2) **Time per training step**. The pattern mirrors memory usage for the same reason: VA-VAE incurs higher cost than our Stage 1 and 3, and our Stage 2 incurs the highest cost when the encoder is trainable.

(3) **Total A100 GPU hours.** The cumulative GPU training hours of our method (107.2 + 377.4 + 91.55 = 576.15) are lower than those of both Vanilla VAE and VA-VAE, demonstrating the overall training efficiency of our pipeline.

We also show cost comparison of inference in Tab. 7 and Tab. 8. Our main observations are as follows:

(1) **Peak GPU Memory.** At lower resolutions and smaller batch sizes, our encoder consumes more peak GPU memory than VA-VAE (ours: 1.372 GB, VA-VAE: 0.838 GB at 256 resolution with a

Table 6: **Training Cost Comparison.** All measurements are obtained at an input resolution of 256×256 on 8 NVIDIA A100 GPUs (batch size 8 per GPU). Values marked with ∗ denote GPU-hour estimates computed from the training epochs reported in the official implementation of (Yao et al., 2025).

| Tokenizer | Enc. #Param | Trainable Params | Frozen Params | Total GPU Memory | Time Per Training Step | Total A100 GPU Hours |
|---|---|---|---|---|---|---|
| Vanilla VAE | 28.41 M | 72.60 M | 14.71 M | 148.2 GB | 0.360 s | 800.0∗ |
| VA-VAE | | 72.63 M | 319.0 M | 203.0 GB | 0.534 s | 2966∗ |
| Our Stage 1 | | 44.25 M | 319.0 M | 157.5 GB | 0.402 s | 107.2 |
| Our Stage 2 | 304.4 M | 348.6 M | 319.0 M | 238.2 GB | 0.772 s | 377.4 |
| Our Stage 3 | | 44.18 M | 319.1 M | 159.0 GB | 0.412 s | 91.55 |

Table 7: **Inference Memory Consumption Comparison.** We evaluate the peak GPU memory consumption under different image resolutions and batch sizes.

| Tokenizer | Image Resolution | Encoder #Params | Encoder Peak GPU Memory (GB) | | | Decoder #Params | Decoder Peak GPU Memory (GB) | | |
|---|---|---|---|---|---|---|---|---|---|
| | | | batch size=1 | batch size=4 | batch size = 8 | | batch size=1 | batch size=4 | batch size = 8 |
| VA-VAE | 256 × 256 | 28.41 M | 0.301 | 0.838 | 1.543 | 41.42 M | 0.310 | 0.713 | 1.250 |
| Ours | 256 × 256 | 304.4 M | 1.363 | 1.372 | 1.385 | | | | |
| VA-VAE | 512 × 512 | 28.41 M | 0.841 | 2.959 | 5.794 | 41.42 M | 0.713 | 2.324 | 4.472 |
| Ours | 512 × 512 | 304.4 M | 1.284 | 1.450 | 1.674 | | | | |
| VA-VAE | 1024 × 1024 | 28.41 M | 2.958 | 11.45 | 22.78 | 41.42 M | 2.324 | 8.768 | 17.36 |
| Ours | 1024 × 1024 | 304.4 M | 1.450 | 2.121 | 3.015 | | | | |

Table 8: **Inference Compute Cost Comparison.** We measure the GFLOPS and Latency under different image resolutions, using a single NVIDIA A100 GPU with a batch size of 1.

| Tokenizer | Image Resolution | Encoder | | | Decoder | | |
|---|---|---|---|---|---|---|---|
| | | #Params | GFLOPS | Latency | #Params | GFLOPS | Latency |
| VA-VAE | 256 × 256 | 28.41 M | 69.21 | 6.548 ms | 41.42 M | 126.5 | 13.91 ms |
| Ours | 256 × 256 | 304.4 M | 77.85 | 13.42 ms | | | |
| VA-VAE | 512 × 512 | 28.41 M | 279.2 | 19.44 ms | 41.42 M | 509.3 | 42.31 ms |
| Ours | 512 × 512 | 304.4 M | 310.4 | 18.92 ms | | | |
| VA-VAE | 1024 × 1024 | 28.41 M | 1155 | 73.92 ms | 41.42 M | 2089 | 173.9 ms |
| Ours | 1024 × 1024 | 304.4 M | 1241 | 121.2 ms | | | |

batch size of 4). However, at higher resolutions and larger batch sizes, our encoder uses substantially less memory (ours: 3.015 GB, VA-VAE: 22.78 GB at 1024 resolution with a batch size of 8).

(2) **Latency and GFLOPS.** Overall, our tokenizer has higher encoding latency and GFLOPS than VA-VAE. The only exception is at 512 resolution, where our tokenizer achieves lower encoding latency.

## C  MORE EXPERIMENTS

### C.1  OTHER PRETRAINED ENCODERS

Tab. 9 reports additional reconstruction metrics comparing different pretrained encoders. The one using DINOv2 achieves the best balance, delivering superior generation quality while maintaining competitive reconstruction performance.

### C.2  RECONSTRUCTION

Fig. 9 shows a qualitative comparison of reconstruction quality on the ImageNet 256×256 dataset. The variant *Ours w/o Stage 2 + 3* (fourth column) fails to reconstruct the input accurately, while the other methods show similar reconstruction quality.

Tab.10 reports additional reconstruction metrics for different tokenizers. While *Ours* (the version presented in the main paper) outperforms all baselines in generative performance, it remains quantitatively weaker in reconstruction quality. To address this, we report a variant of our method trained with a larger learning rate (increased from 1e-5 to 1e-4) and a smaller semantic preservation weight

Table 9: **Comparison with Other Pretrained Encoders.** Evaluated on ImageNet 256×256 at 80K training steps with 30 sampling steps, using the CFG scale that yields the lowest gFID. Decoder refinements (stage 3) are not applied. While the model with MAE yields the strongest reconstruction performance, it performs the worst in generation quality. The model with DINOv2 achieves the best overall generation quality, with a slight drop in reconstruction quality compared to MAE.

| Configuration | rFID↓ | PSNR↑ | LPIPS↓ | gFID↓ | IS↑ | Prec↑ | Recall↑ |
|---|---|---|---|---|---|---|---|
| MAE | **0.29** | **26.12** | **0.113** | 3.12 | 216.5 | **0.834** | 0.543 |
| SigLIP 2 | 0.35 | 25.29 | 0.129 | 2.22 | 246.1 | 0.816 | 0.576 |
| DINOv2 | 0.36 | 25.62 | 0.121 | **2.19** | **248.6** | 0.811 | **0.591** |

Table 10: **Comparison of Other Tokenizers with Different Configurations.** Evaluated on ImageNet 256×256 at 80K training steps with 30 sampling steps. The checkpoints for both the Vanilla VAE and VA-VAE with CNN encoders are taken from the official VA-VAE repository. *VA-VAE†* denotes the VA-VAE model we trained, using a ViT encoder that matches our architecture but initialized from scratch. *Ours* refers to the version of our method presented in the main paper. *Ours\** indicates a variant trained with a larger learning rate and smaller semantic preservation weight, which achieves reconstruction quality comparable to VA-VAE but with slightly reduced generation performance.

| Tokenizer | Enc. Arch. | rFID↓ | PSNR↑ | LPIPS↓ | L. P. Acc.↑ | gFID w/ CFG↓ |
|---|---|---|---|---|---|---|
| *f16d32 (downsampling factor 16, latent dimension 32)* | | | | | | |
| Vanilla VAE | CNN | **0.26** | **27.14** | **0.097** | 6.04% | 3.31 |
| VA-VAE | CNN | 0.28 | 26.31 | 0.104 | 22.96% | 3.13 |
| VA-VAE† | ViT | 0.37 | 25.66 | 0.130 | 33.57% | 3.16 |
| Ours | ViT | **0.26** | 25.83 | 0.117 | **35.09%** | **2.17** |
| *f16d64 (downsampling factor 16, latent dimension 64)* | | | | | | |
| Vanilla VAE | CNN | 0.17 | **29.38** | **0.061** | 5.09% | 4.03 |
| VA-VAE | CNN | **0.14** | 29.13 | 0.062 | 19.72% | 3.20 |
| VA-VAE† | ViT | 0.18 | 29.12 | 0.075 | 43.53% | 3.19 |
| Ours | ViT | 0.17 | 27.41 | 0.089 | **46.99%** | **2.34** |
| Ours\* | ViT | **0.14** | 28.91 | 0.070 | 45.22% | 2.50 |

(reduced from 1.0 to 0.5) in stage 2. This hyperparameter setting pushes the tokenizer to learn perceptual details more aggressively. For this variant, we train the first stage for 6 epochs, second stage for 11 epochs, the final stage for 10 epochs. The resulting variant achieves competitive reconstruction performance (on par with VA-VAE using ViT encoder) while still surpassing VA-VAE in generation. This demonstrates that our method can improve reconstruction with only a minor trade-off in generative quality through simple hyperparameter adjustments. Other strategies for improvement include extending stage 2 or stage 3 with additional training iterations, or using a larger batch size. A more aggressive approach is to replace the decoder with a stronger architecture during stage 3 and train the new decoder. While this requires additional training resources, it offers flexibility without compromising the learned semantic space.

## C.3 CONVERGENCE SPEED

Figs. 10, Fig. 11, and Fig. 12 present additional comparisons between our method and VA-VAE on the ImageNet 256×256 dataset. Our method converges faster, indicating that aligning and preserving semantics in the pretrained encoder is more effective than semantic regularization.

Fig. 13, Fig. 14, Fig. 15, and Fig. 16 compare the convergence speed of our method with FLUX VAE. Our method converges significantly faster, demonstrating the advantage of the learned semantically rich latent space.

## C.4 MORE QUANTITATIVE RESULTS

Tab. 11 reports quantitative results on two additional prompt sets. Tab. 13 and Tab. 15 present comparisons against FLUX VAE and VA-VAE on GenEval Ghosh et al. (2023). Tab. 12 and Tab. 14

Table 11: **Quantitative Comparison on Text-to-Image (T2I) Generation with FLUX VAE.** We report metrics on two additional prompt sets for T2I models trained with our VAE and FLUX VAE, each for 100K steps, evaluated at 256×256 resolution. The rFID is computed using 200K randomly sampled images from the COYO-700M dataset (Minwoo et al., 2022).

| Tokenizer | rFID | HPSv2 | PickScore | ImageReward | Aesthetic Scores | CLIP Scores | VQA Scores |
|---|---|---|---|---|---|---|---|
| *Parti Prompt (Yu et al., 2022) (with CFG)* | | | | | | | |
| FLUX VAE | **0.102** | 0.239 | 0.391 | 0.235 | 5.292 | 31.49 | 0.705 |
| Ours | 0.443 | **0.246** | **0.609** | **0.594** | **5.389** | **32.56** | **0.782** |
| *HPSv2 Prompt (Wu et al., 2023) (with CFG)* | | | | | | | |
| FLUX VAE | **0.102** | 0.224 | 0.373 | 0.090 | 5.478 | 31.59 | 0.737 |
| Ours | 0.443 | **0.231** | **0.626** | **0.366** | **5.604** | **33.12** | **0.792** |

Table 12: **Quantitative Comparison on Text-to-Image (T2I) Generation with FLUX VAE (Without CFG).** We report metrics on diverse prompt sets for 2B-parameter T2I models trained with our tokenizer and the FLUX VAE. Each T2I model is trained for 100K steps and evaluated at 256 × 256 resolution. Our tokenizer is trained on LAION dataset. The rFID is computed using 200K randomly sampled images from the COYO-700M dataset (Minwoo et al., 2022).

| Tokenizer | rFID | gFID | KID | HPSv2 | PickScore | ImageReward | Aesthetic Scores | CLIP Scores | VQA Scores |
|---|---|---|---|---|---|---|---|---|---|
| *Coco Prompt 6K (Lin et al., 2014) (without CFG)* | | | | | | | | | |
| FLUX VAE | **0.102** | 48.76 | 0.025 | 0.183 | 0.345 | -1.127 | 4.205 | 27.05 | 0.534 |
| Ours | 0.443 | **33.46** | **0.014** | **0.210** | **0.655** | **-0.581** | **4.963** | **28.87** | **0.658** |
| *Parti Prompt (Yu et al., 2022) (without CFG)* | | | | | | | | | |
| FLUX VAE | **0.102** | - | - | 0.186 | 0.378 | -1.176 | 4.003 | 26.85 | 0.538 |
| Ours | **0.443** | - | - | **0.206** | **0.622** | **-0.654** | **4.751** | **28.59** | **0.618** |
| *HPSv2 Prompt (Wu et al., 2023) (without CFG)* | | | | | | | | | |
| FLUX VAE | **0.102** | - | - | 0.170 | 0.372 | −1.206 | 4.142 | 26.59 | 0.567 |
| Ours | 0.443 | - | - | **0.192** | **0.628** | **-0.764** | **4.955** | **28.20** | **0.645** |

Table 13: **GenEval Quanlitative Comparison on Text-to-Image (T2I) Generation with FLUX VAE (with and without CFG).** We report metrics on GenEval (Ghosh et al., 2023) for 2B-parameter T2I models trained with our tokenizer and FLUX VAE. Each T2I model is trained for 100K steps and evaluated at 256×256 resolution. Our tokenizer is trained on LAION dataset. The rFID is computed using 200K randomly sampled images from the COYO-700M dataset (Minwoo et al., 2022).

| Tokenizer | rFID | Overall | Single object | Two object | Counting | Colors | Position | Color attribution |
|---|---|---|---|---|---|---|---|---|
| *with CFG* | | | | | | | | |
| FLUX VAE | **0.102** | 0.476 | 0.978 | 0.477 | 0.346 | 0.813 | 0.107 | 0.135 |
| Ours | 0.443 | **0.556** | **0.984** | **0.702** | **0.446** | **0.824** | **0.120** | **0.257** |
| *without CFG* | | | | | | | | |
| FLUX VAE | **0.102** | 0.230 | 0.746 | 0.101 | 0.090 | 0.406 | 0.020 | 0.015 |
| Ours | 0.443 | **0.329** | **0.850** | **0.303** | **0.234** | **0.508** | **0.037** | **0.040** |

further compare the main metrics under settings with and without CFG. Across all configurations, our tokenizer consistently outperforms both FLUX VAE and VA-VAE in generation quality, regardless of whether CFG is applied.

## C.5 MORE QUALITATIVE RESULTS

Fig. 17 and Fig. 18 show the comparison with FLUX-VAE (withou CFG) and VA-VAE (with CFG).

Fig. 19 shows our sampled results of class-conditioned image generation on ImageNet 256×256 dataset, trained with 800 epochs.

Fig. 20 shows our sampled results of text-to-image generation at resolution 256×256, trained with 200K steps.

Fig. 21 and Fig. 22 show our sampled results of text-to-image generation at resolution 512×512. Fig. 23 and Fig. 24 further show results at different aspect ratios, including portrait mode and landscape mode. All these results are generated using the model trained with 290K steps.

Table 14: **Quantitative Comparison on Text-to-Image (T2I) Generation with VA-VAE (with and without CFG).** We report metrics on diverse prompt sets for 1B-parameter T2I models trained with our tokenizer and the VA-VAE. Each T2I model is trained for 50K steps and evaluated at 256×256 resolution. Both our tokenizer and VA-VAE are trained on ImageNet. The rFID is computed using 200K randomly sampled images from the COYO-700M dataset (Minwoo et al., 2022).

| Tokenizer | rFID | gFID | KID | HPSv2 | PickScore | ImageReward | Aesthetic Scores | CLIP Scores | VQA Scores |
|---|---|---|---|---|---|---|---|---|---|
| | | | | *Coco Prompt 6K (Lin et al., 2014) (with CFG)* | | | | | |
| VA-VAE | 0.241 | 34.13 | 0.016 | 0.221 | 0.426 | -0.129 | 5.088 | 31.50 | 0.740 |
| Ours | **0.191** | **31.19** | **0.015** | **0.230** | **0.574** | **0.088** | **5.215** | **31.68** | **0.781** |
| | | | | *Parti Prompt (Yu et al., 2022) (with CFG)* | | | | | |
| VA-VAE | 0.241 | - | - | 0.220 | 0.444 | -0.146 | 4.893 | 31.40 | 0.677 |
| Ours | **0.191** | - | - | **0.228** | **0.556** | **0.062** | **5.015** | **31.73** | **0.706** |
| | | | | *HPSv2 Prompt (Wu et al., 2023) (with CFG)* | | | | | |
| VA-VAE | 0.241 | - | - | 0.201 | 0.435 | -0.287 | 5.042 | 31.63 | 0.708 |
| Ours | **0.191** | - | - | **0.210** | **0.565** | **-0.131** | **5.120** | **32.05** | **0.737** |
| | | | | *Coco Prompt 6K (Lin et al., 2014) (without CFG)* | | | | | |
| VA-VAE | 0.241 | 42.35 | 0.019 | 0.179 | 0.448 | -1.292 | 4.352 | 26.89 | 0.497 |
| Ours | **0.191** | **39.86** | **0.017** | **0.187** | **0.552** | **-1.121** | **4.493** | **27.53** | **0.548** |
| | | | | *Parti Prompt (Yu et al., 2022) (without CFG)* | | | | | |
| VA-VAE | 0.241 | - | - | 0.179 | 0.457 | -1.396 | 4.163 | 26.23 | 0.493 |
| Ours | **0.191** | - | - | **0.186** | **0.543** | **-1.213** | **4.320** | **26.99** | **0.524** |
| | | | | *HPSv2 Prompt (Wu et al., 2023) (without CFG)* | | | | | |
| VA-VAE | 0.241 | - | - | 0.162 | 0.455 | -1.378 | 4.256 | 26.05 | 0.521 |
| Ours | **0.191** | - | - | **0.170** | **0.545** | **-1.251** | **4.451** | **26.48** | **0.551** |

Table 15: **GenEval Quanlitative Comparison on Text-to-Image (T2I) Generation with VA-VAE (with and without CFG).** We report metrics on GenEval (Ghosh et al., 2023) for 1B-parameter T2I models trained with our tokenizer and VA-VAE. Each T2I model is trained for 50K steps and evaluated at 256×256 resolution. Both our tokenizer and VA-VAE are trained on ImageNet. The rFID is computed using 200K randomly sampled images from the COYO-700M dataset (Minwoo et al., 2022).

| Tokenizer | rFID | Overall | Single object | Two object | Counting | Colors | Position | Color attribution |
|---|---|---|---|---|---|---|---|---|
| | | | | *with CFG* | | | | |
| VA-VAE | 0.241 | 0.411 | 0.944 | 0.338 | 0.275 | 0.782 | 0.050 | 0.078 |
| Ours | **0.191** | **0.454** | **0.978** | **0.434** | **0.338** | **0.795** | **0.073** | **0.110** |
| | | | | *without CFG* | | | | |
| VA-VAE | 0.241 | 0.194 | 0.653 | 0.071 | 0.088 | 0.335 | 0.010 | 0.005 |
| Ours | **0.191** | **0.243** | **0.734** | **0.136** | **0.125** | **0.418** | **0.033** | **0.013** |

Table 16: **Quantitative Analysis of Latent Space.** We report metrics for latent space of three tokenizers (Vanilla VAE, VA-VAE, and Ours), using the output space of the pretrained DINOv2 encoder as a reference. CKNNA Huh et al. (2024) measures the similarity between each tokenizer's latent space and the DINOv2 output. For the other metrics, we additionally report the percentage indicating how close each tokenizer's results are to those of DINOv2. All metrics are computed using 10K samples from the ImageNet validation set.

| Tokenizer | CKNNA | Spatial Variance | Total Variation | Density CV | Gini Coefficient | Normalized Entropy | gFID |
|---|---|---|---|---|---|---|---|
| Vanilla VAE | 0.023 | 11.41 (+432.9%) | 957.6 (+96.3%) | 0.310 (+24.0%) | 0.173 (+24.4%) | 0.994 (-0.2%) | 3.31 |
| VA-VAE | 0.233 | 13.74 (+541.7%) | 806.1 (+65.1%) | **0.215 (-14.0%)** | **0.122 (-12.2%)** | **0.997 (+0.1%)** | 3.13 |
| Ours | **0.282** | **1.205 (-43.7%)** | **302.3 (-38.0%)** | 0.295 (+18.0%) | 0.160 (+15.1%) | 0.994 (-0.2%) | **2.17** |
| DINOv2 | 1.000 | 2.141 | 487.9 | 0.250 | 0.139 | 0.996 | - |

## C.6  ANALYSIS OF THE SEMANTICS OF THE LATENT SPACE

Fig. 7 shows PCA visualizations of the latent space. The figure compares the latent spaces produced by three tokenizers (Vanilla VAE, VA-VAE, and Ours) with the output of the pretrained DINOv2 encoder as a semantic reference. Vanilla VAE tends to produce latents that are either overly smooth or overly sharp, while VA-VAE exhibits oversmoothing. In contrast, our latent space most closely resembles the structure of DINOv2 features, reflecting its stronger semantic structure.

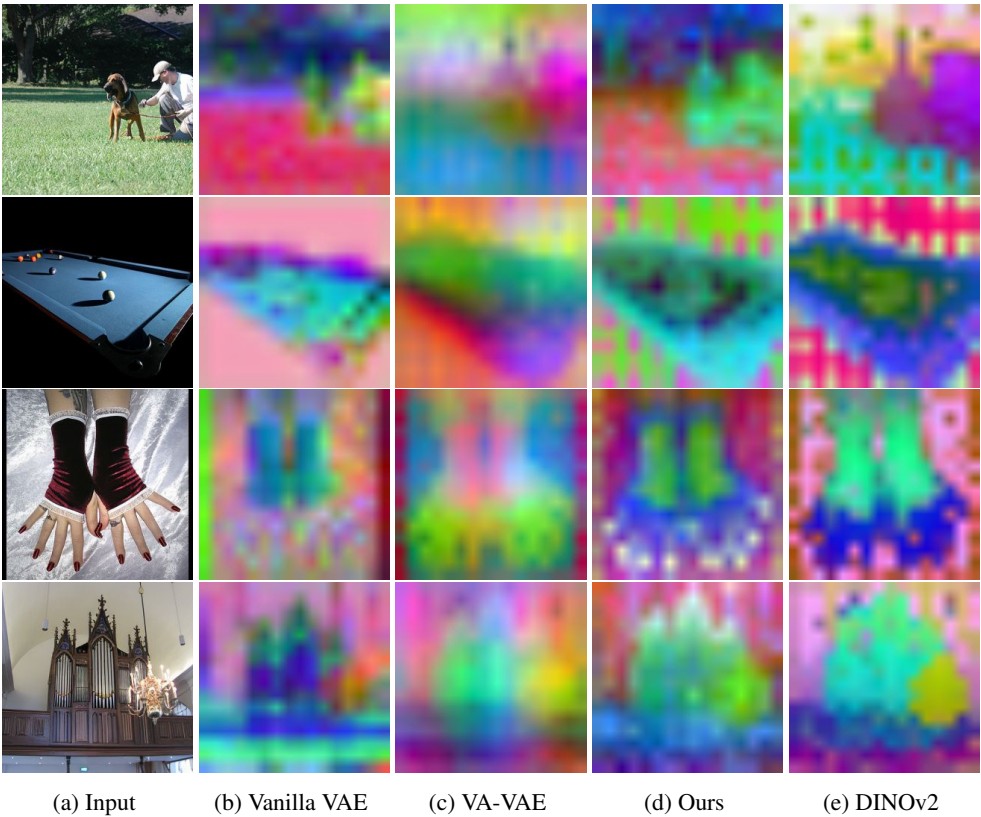

| (a) Input | (b) Vanilla VAE | (c) VA-VAE | (d) Ours | (e) DINOv2 |

Figure 7: **PCA Visualization of Latent Space.** Vanilla VAE can produce latents that are either over-smooth or overly sharp, and VA-VAE tends to generate consistently over-smooth representations. In contrast, our latent space most closely matches the characteristics of DINOv2 outputs, demonstrating that our tokenizer preserves richer semantics.

Besides, Tab. 16 shows the quantitative analysis of the latent space. We evaluate several distributional metrics such as CKNNA Huh et al. (2024), Total Variation, and Gini Coefficient. While our latent space aligns more closely with the DINOv2 space than other tokenizers in metrics like CKNNA and Total Variation, VA-VAE appears closest in terms of metrics like Gini Coefficient. We note, however, that each of these metrics captures only one aspect of the latent distribution, and none of them individually correlates perfectly with semantic quality. This is why we relied primarily on linear probing to evaluate the semantic of the latent space.

## C.7 FAILURE CASE

Fig. 8 shows the failure case of our method. Issues include inaccurate rendering of clock numerals (*e.g.*, the 12), incorrect object counts, incorrect long text generation, and difficulties in rendering fine details such as hands.

## D LLM USAGE

In preparing this manuscript, we used large language models (LLMs) as general-purpose writing assistants for grammar corrections, rephrasing, and clarity/concision edits. All LLM-suggested edits were reviewed and verified by the authors, who take full responsibility for the final manuscript.

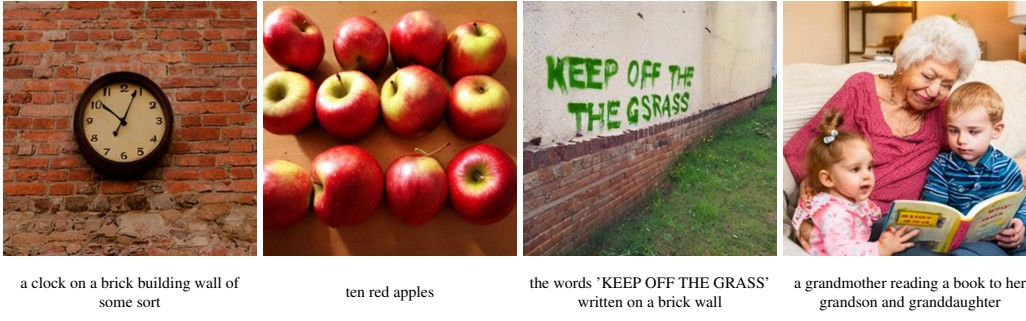


a clock on a brick building wall of some sort     ten red apples     the words 'KEEP OFF THE GRASS' written on a brick wall     a grandmother reading a book to her grandson and granddaughter


Figure 8: **Failure Case of Our Method on Text-to-Image Generation at 512×512 Resolution.** The input text prompts are shown below the images. Results are obtained from generative models trained for 290K steps. Common issues include inaccurate rendering of clock numerals (*e.g.*, the 12), errors in object counting, inconsistencies in generating longer text, and difficulties in producing fine details such as hands.

Input       Vanilla       VA-VAE       Ours w/o Stage 2 + 3       Ours

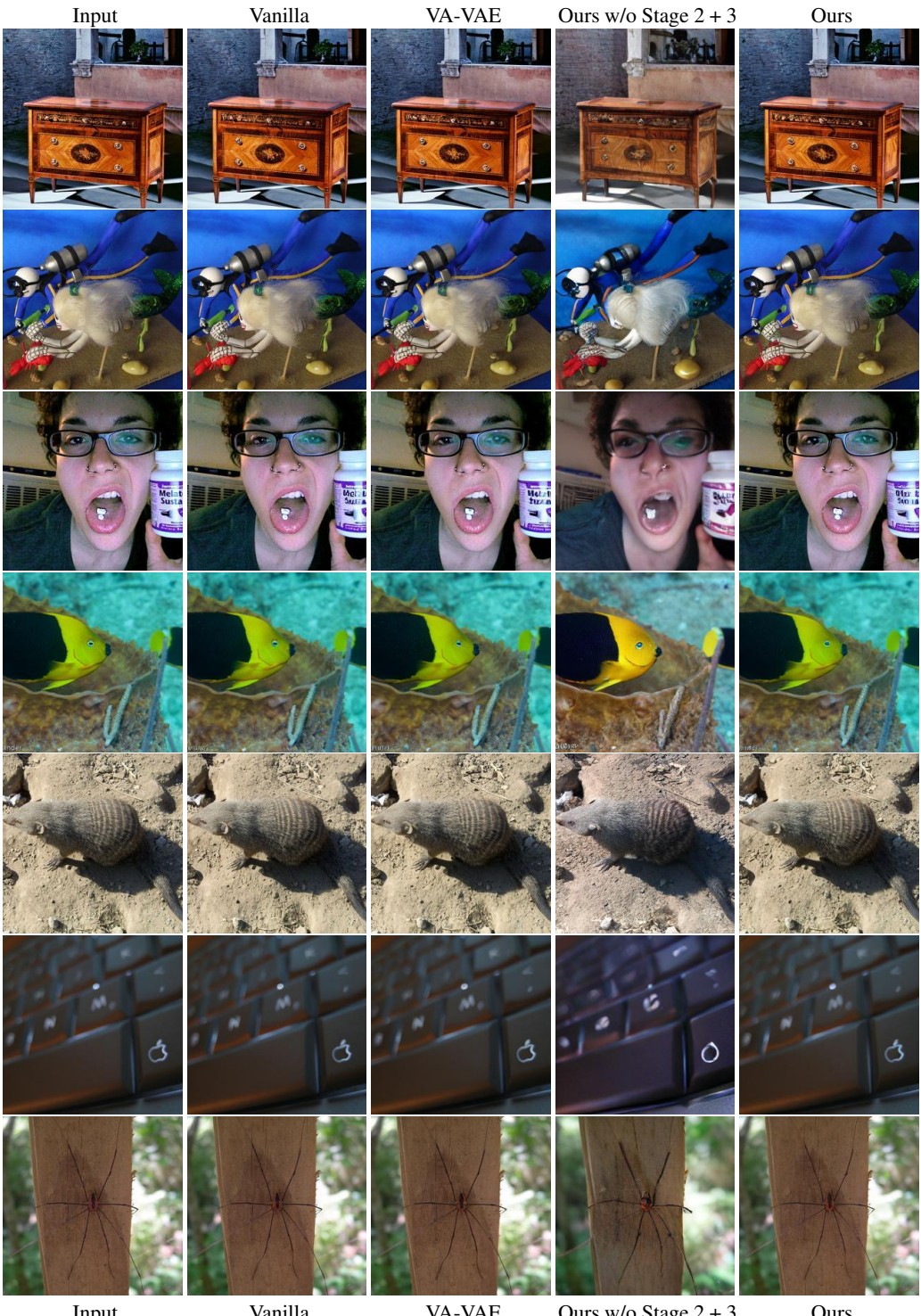

Input       Vanilla       VA-VAE       Ours w/o Stage 2 + 3       Ours

Figure 9: **Qualitative Comparison of Reconstruction Quality on ImageNet 256×256 Validation Set.** The variant *Ours w/o Stage 2 + 3* (fourth column) fails to accurately reconstruct the input, as the pretrained visual encoder does not capture sufficient perceptual details in the latent space. All the other methods achieve comparable reconstruction quality qualitatively.

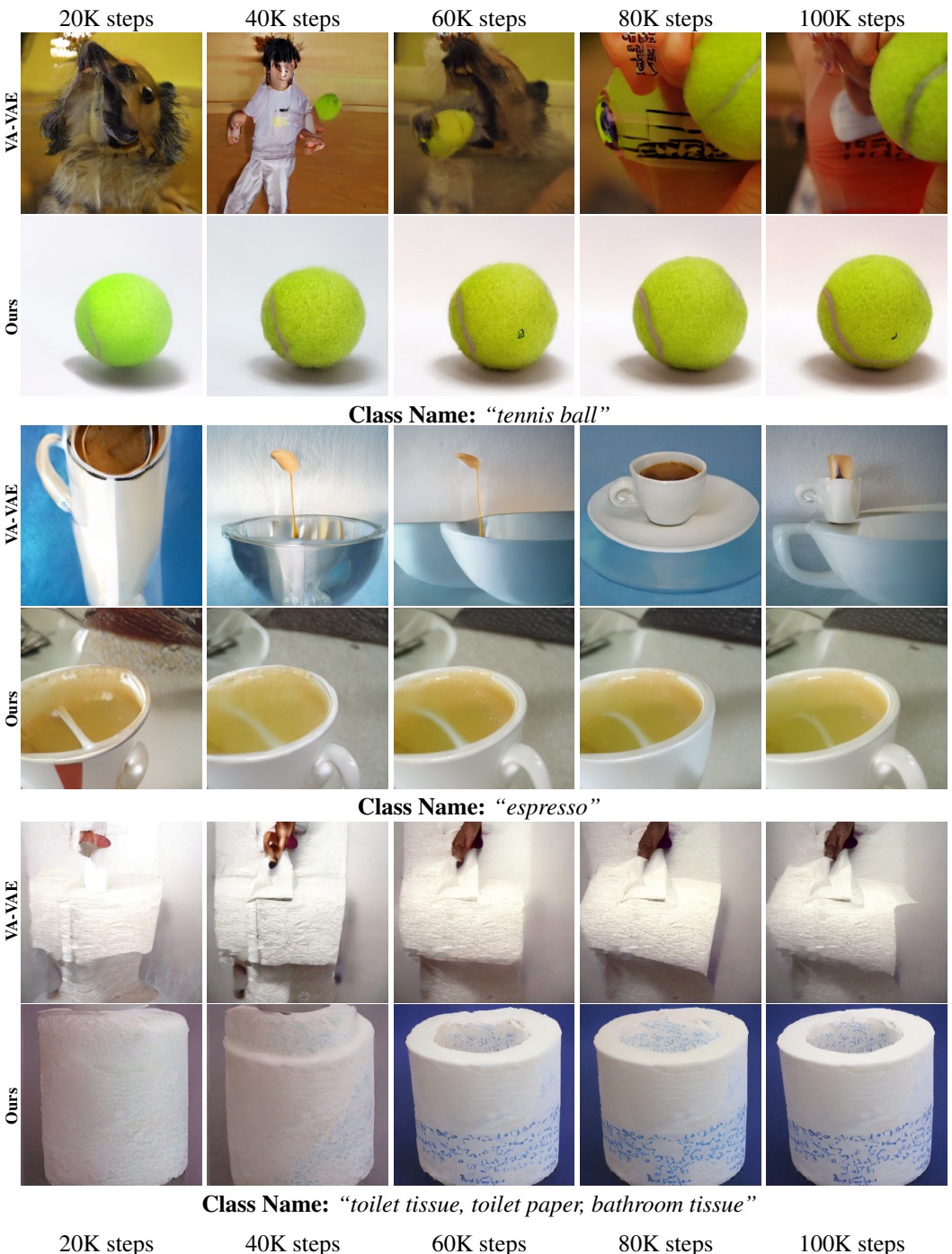

Figure 10: **Qualitative Comparison of Convergence Speed on ImageNet 256×256.** We compare with VA-VAE. Results are reported with the best CFG scale, using EMA for sampling except at 20K and 40K steps.

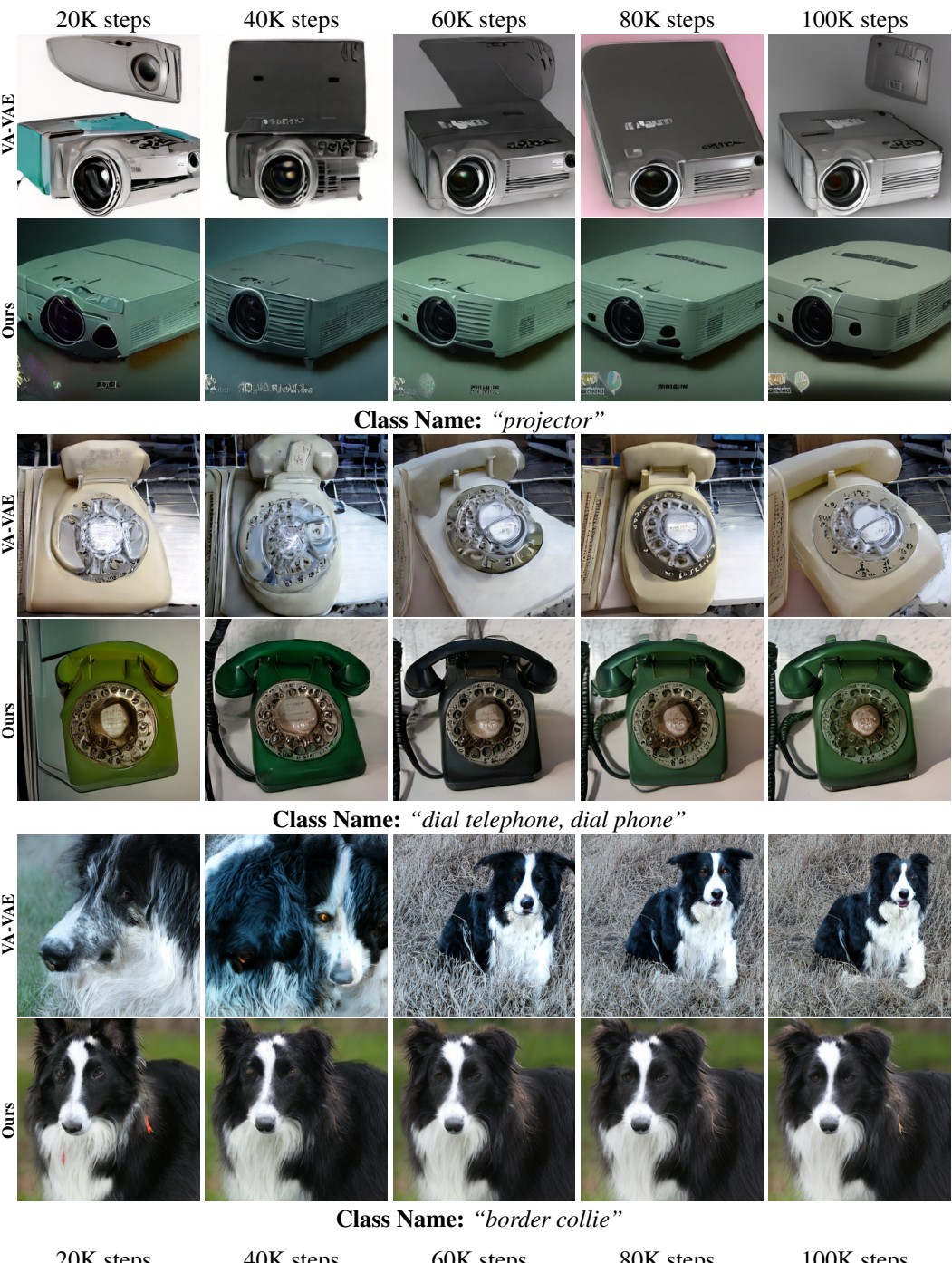

Figure 11: **Qualitative Comparison of Convergence Speed on ImageNet 256×256.** We compare with VA-VAE. Results are reported with the best CFG scale, using EMA for sampling except at 20K and 40K steps.

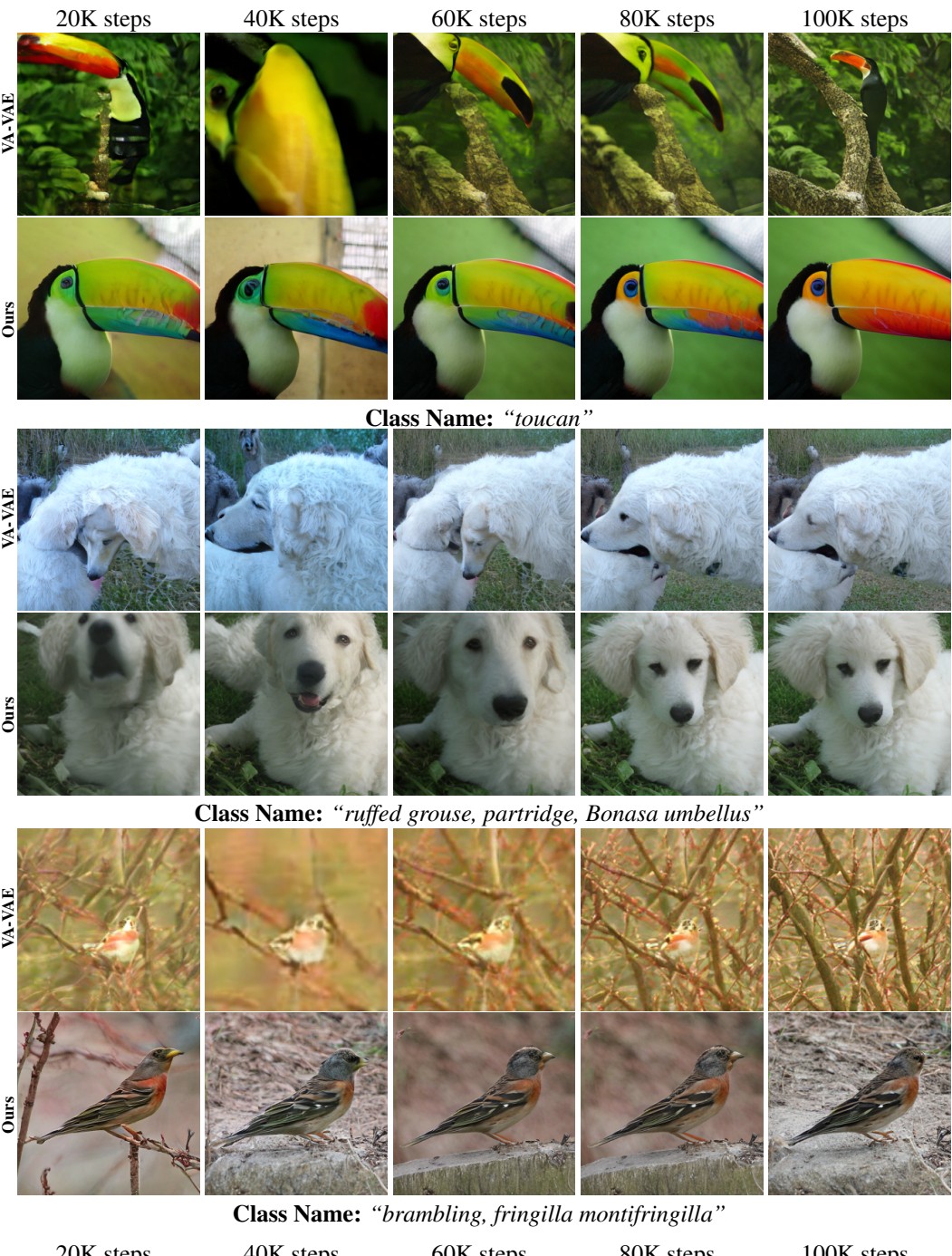

Figure 12: **Qualitative Comparison of Convergence Speed on ImageNet 256×256.** We compare with VA-VAE. Results are reported with the best CFG scale, using EMA for sampling except at 20K and 40K steps.

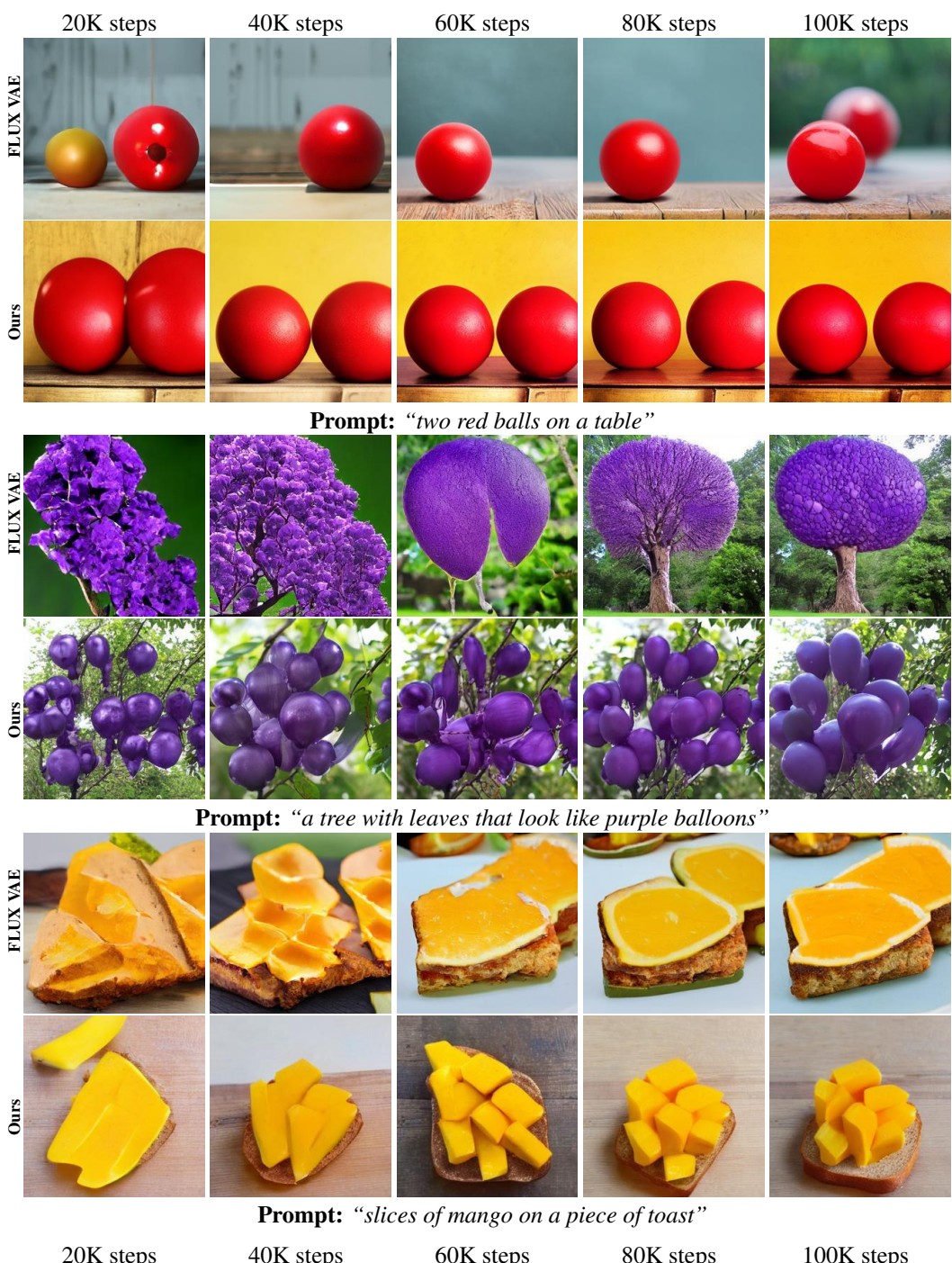

Figure 13: **Qualitative Comparison of Convergence Speed on Text-to-Image Generation.** We compare with FLUX VAE. Results are reported with CFG scale set to 5, using images generated at 256×256 resolution.

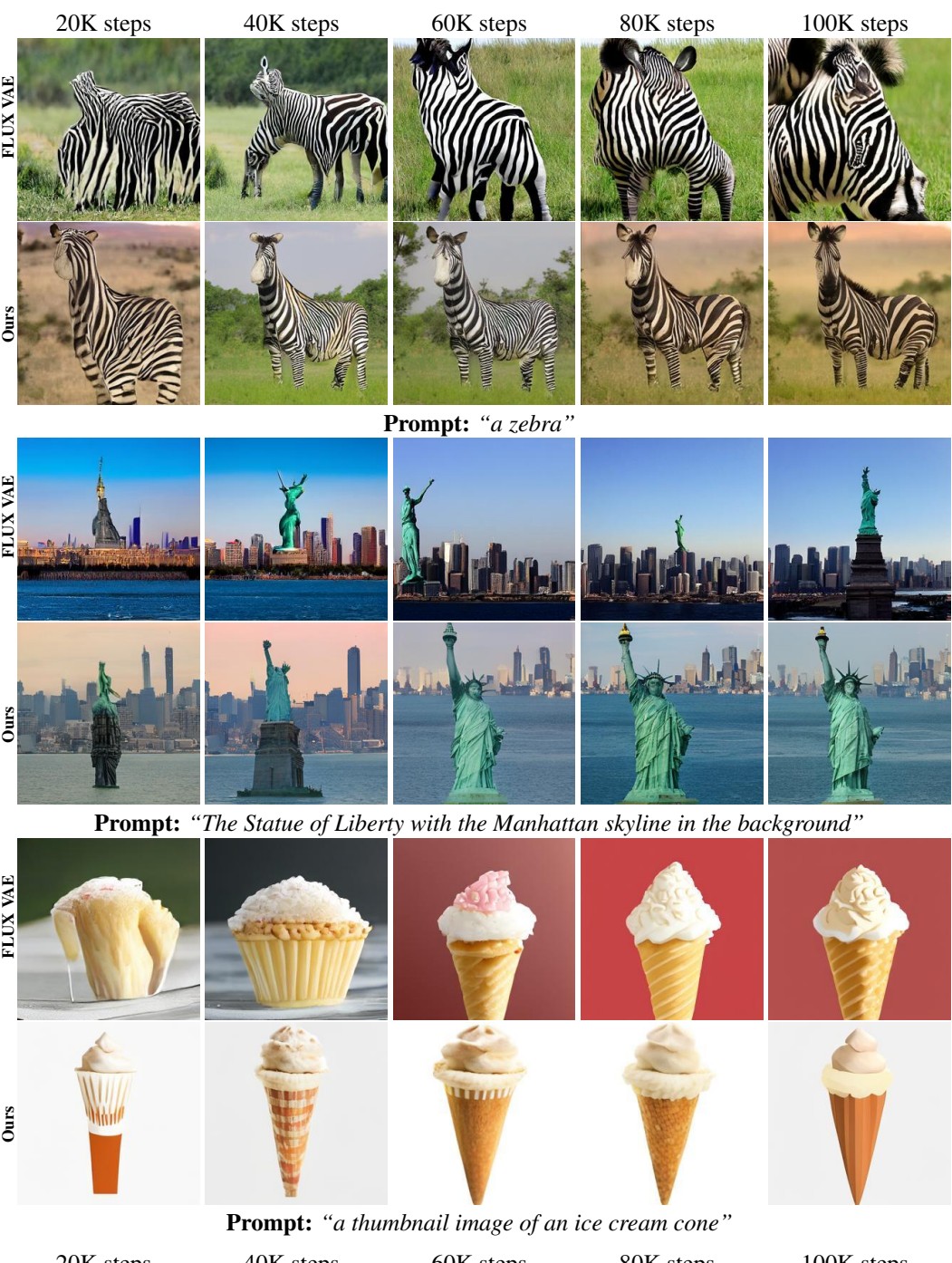

Figure 14: **Qualitative Comparison of Convergence Speed on Text-to-Image Generation.** We compare with FLUX VAE. Results are reported with CFG scale set to 5, using images generated at 256×256 resolution.

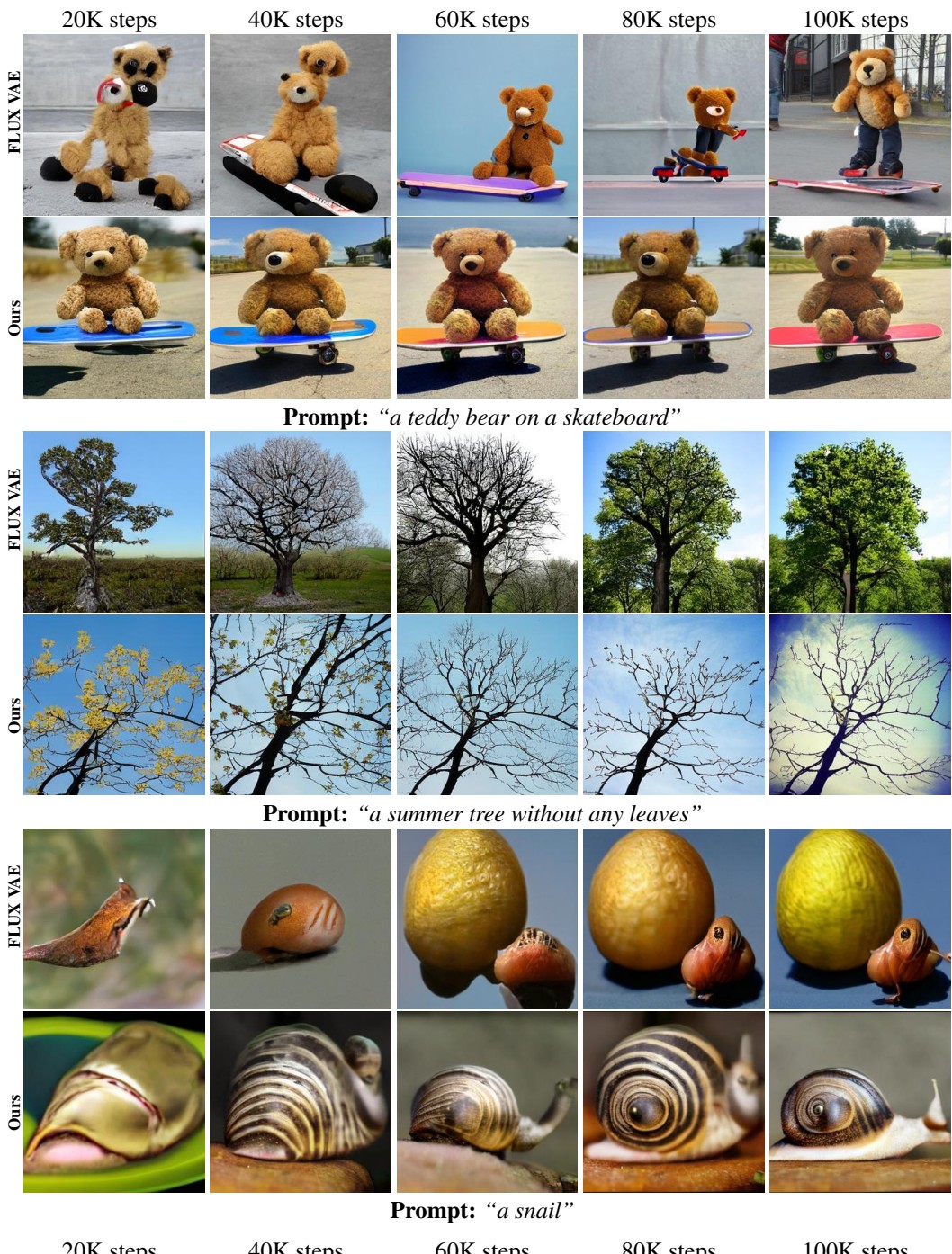

Figure 15: **Qualitative Comparison of Convergence Speed on Text-to-Image Generation.** We compare with FLUX VAE. Results are reported with CFG scale set to 5, using images generated at 256×256 resolution.

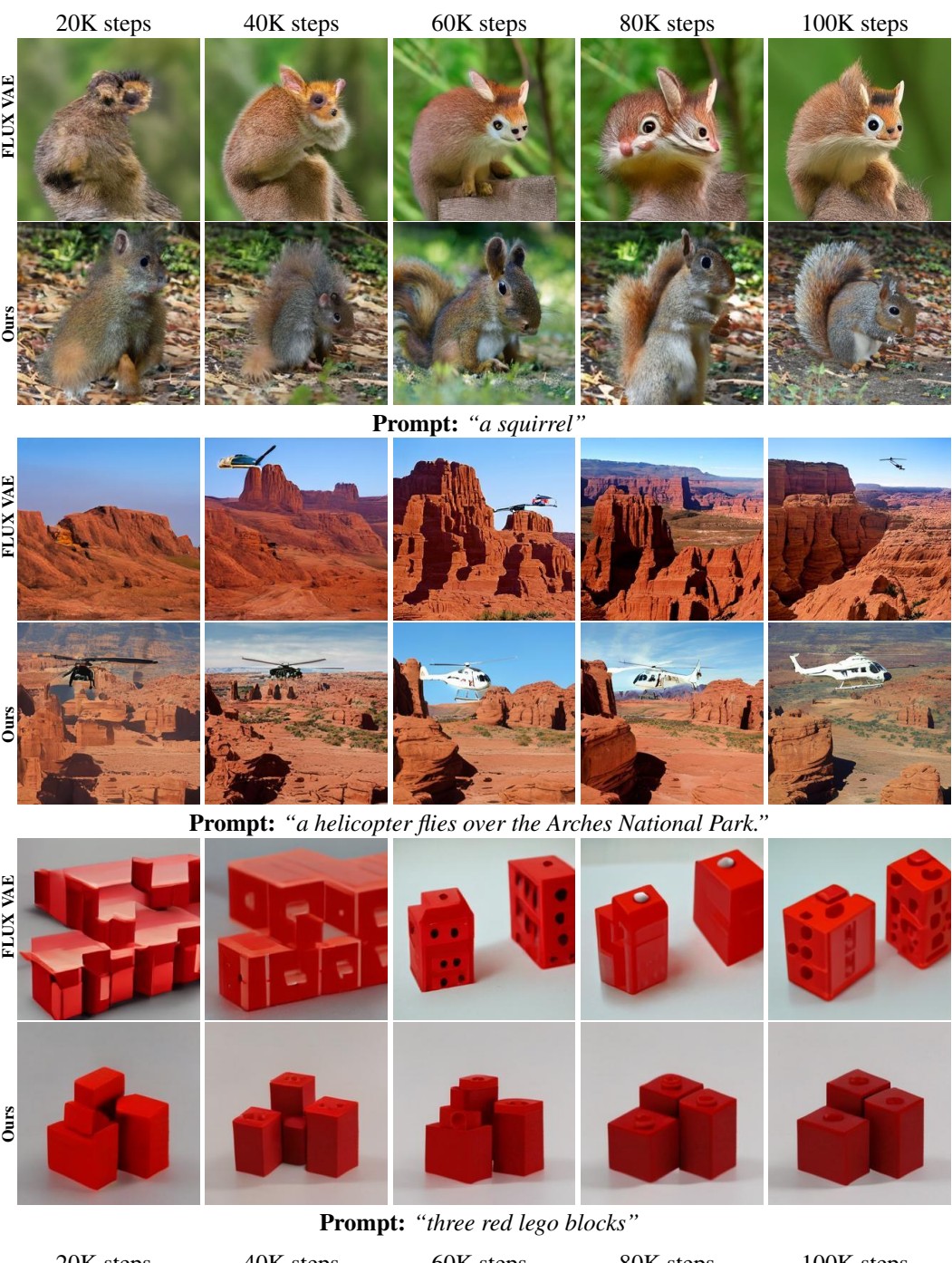

Figure 16: **Qualitative Comparison of Convergence Speed on Text-to-Image Generation.** We compare with FLUX VAE. Results are reported with CFG scale set to 5, using images generated at 256×256 resolution.

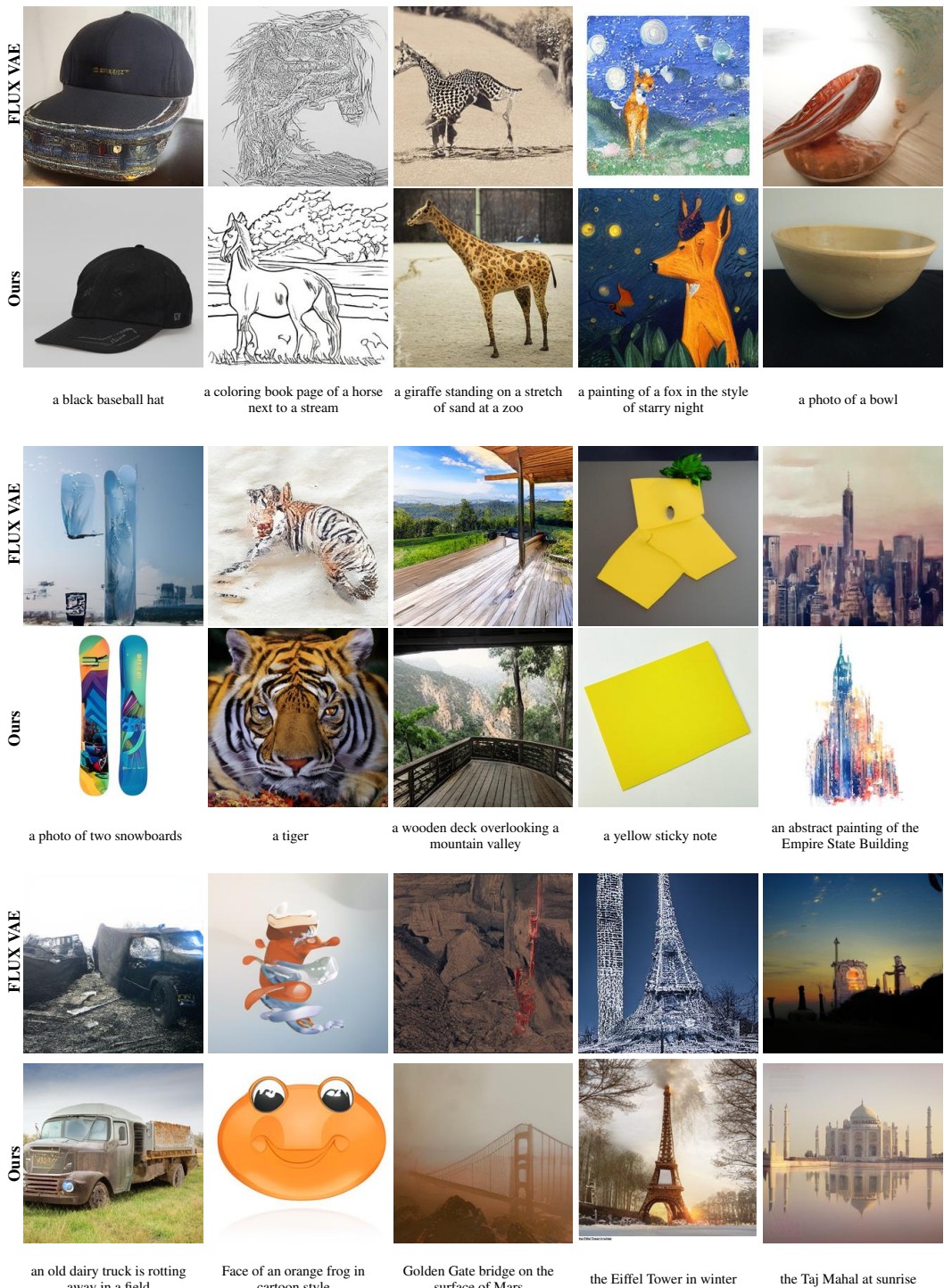

Figure 17: **Qualitative Comparison on Text-to-Image Generation with FLUX VAE (without CFG).** Input text prompts are shown below the images and results (256×256 resolution) are generated from 2B-parameter diffusion models trained for 100K steps, without using CFG. Our method (bottom row) produces images with better coherence and prompt alignment compared to the one using FLUX VAE (top row).

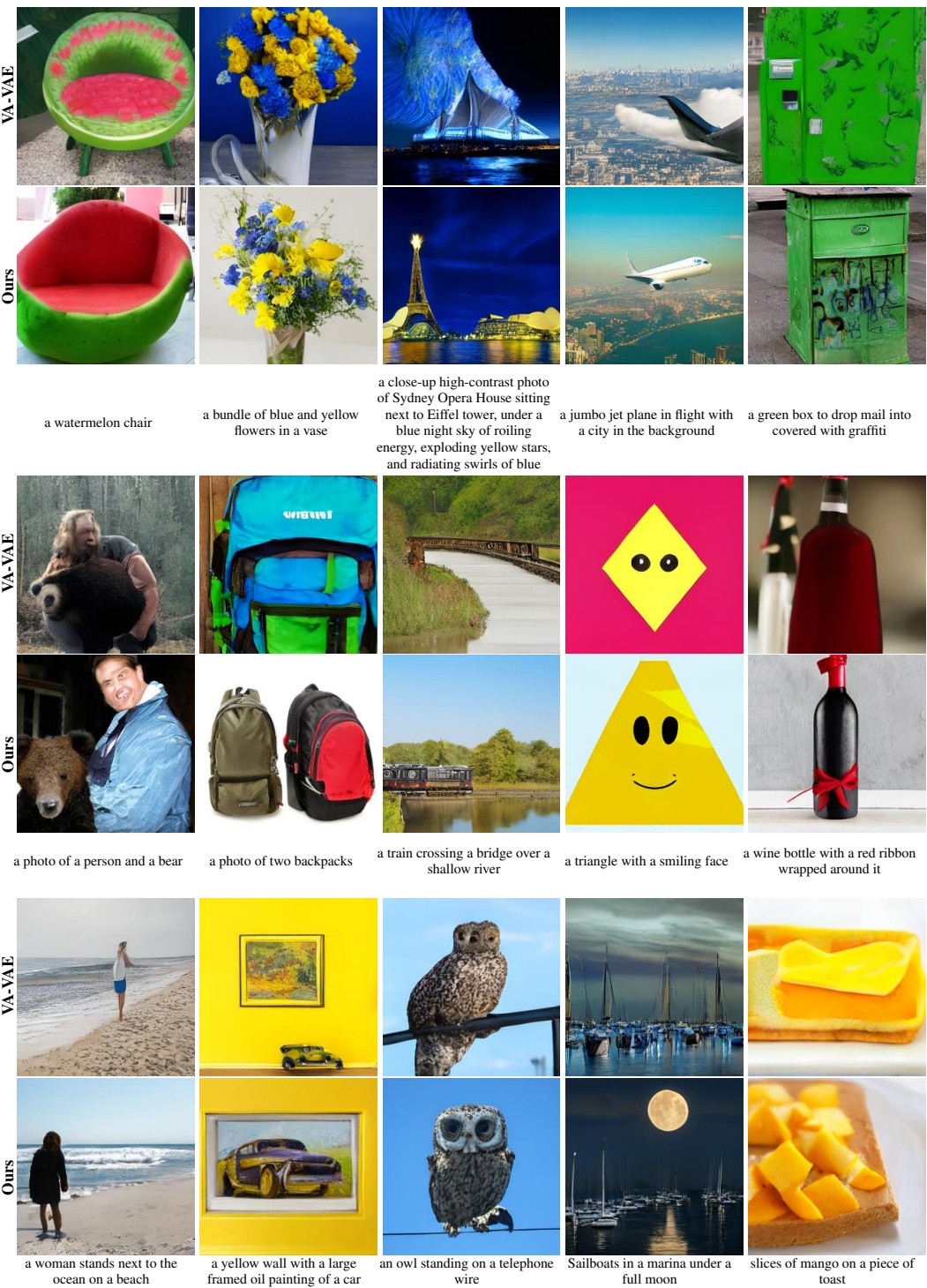

Figure 18: **Qualitative Comparison on Text-to-Image Generation with VA-VAE.** Input text prompts are shown below the images and results (256×256 resolution) are generated from 1B-parameter diffusion models trained for 50K steps, using CFG scale of 5. Note that, both our tokenizer and VA-VAE are trained on ImageNet. Our method (bottom row) produces images with better coherence and prompt alignment compared to the one using VA-VAE (top row).

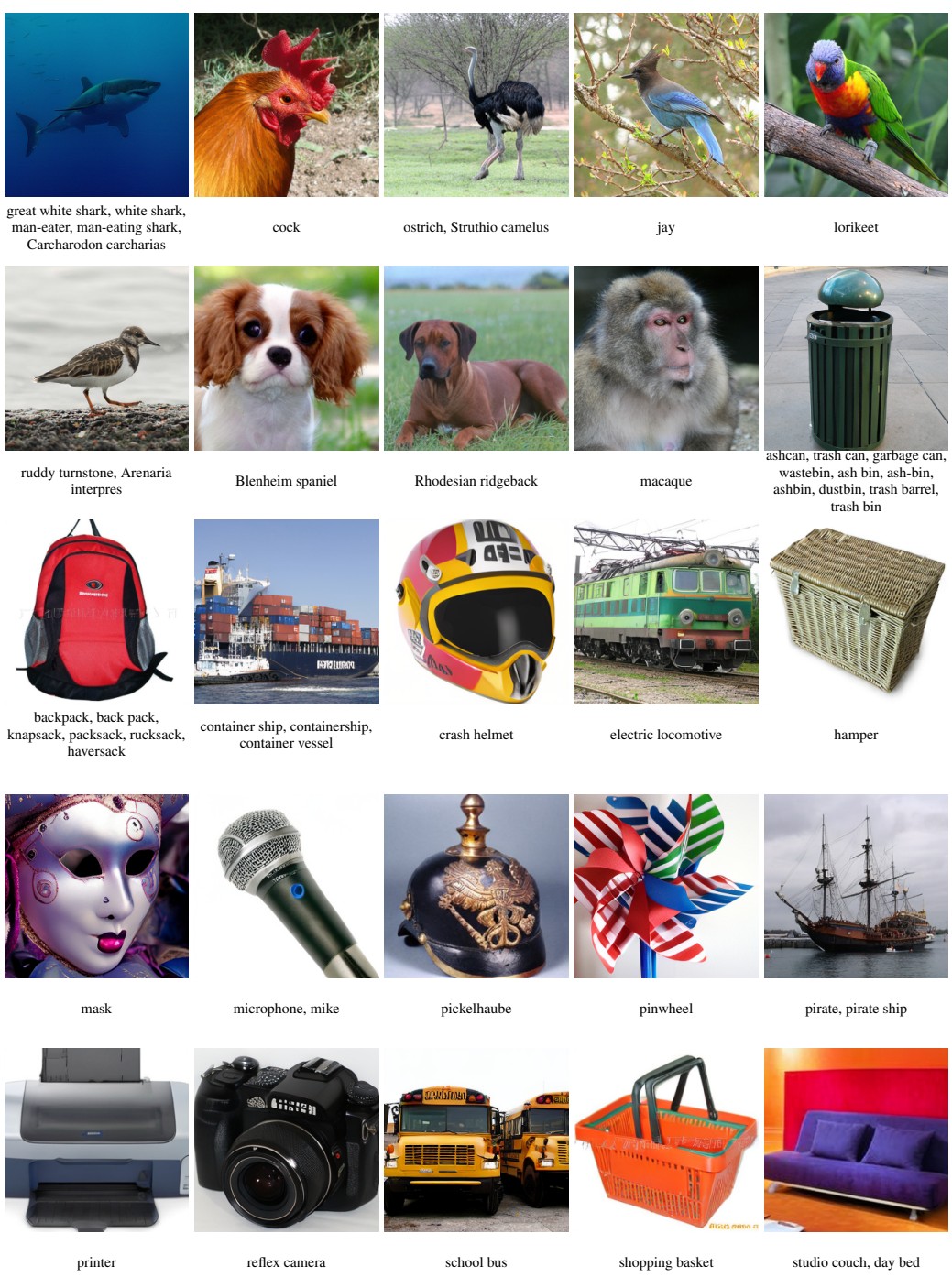

Figure 19: **Qualitative Results of Our Method on ImageNet Class-Conditional Generation.** ImageNet class names are shown below the images and results are from diffusion models trained with 800 epochs. We set the CFG to 5 and apply it to all latent channels.

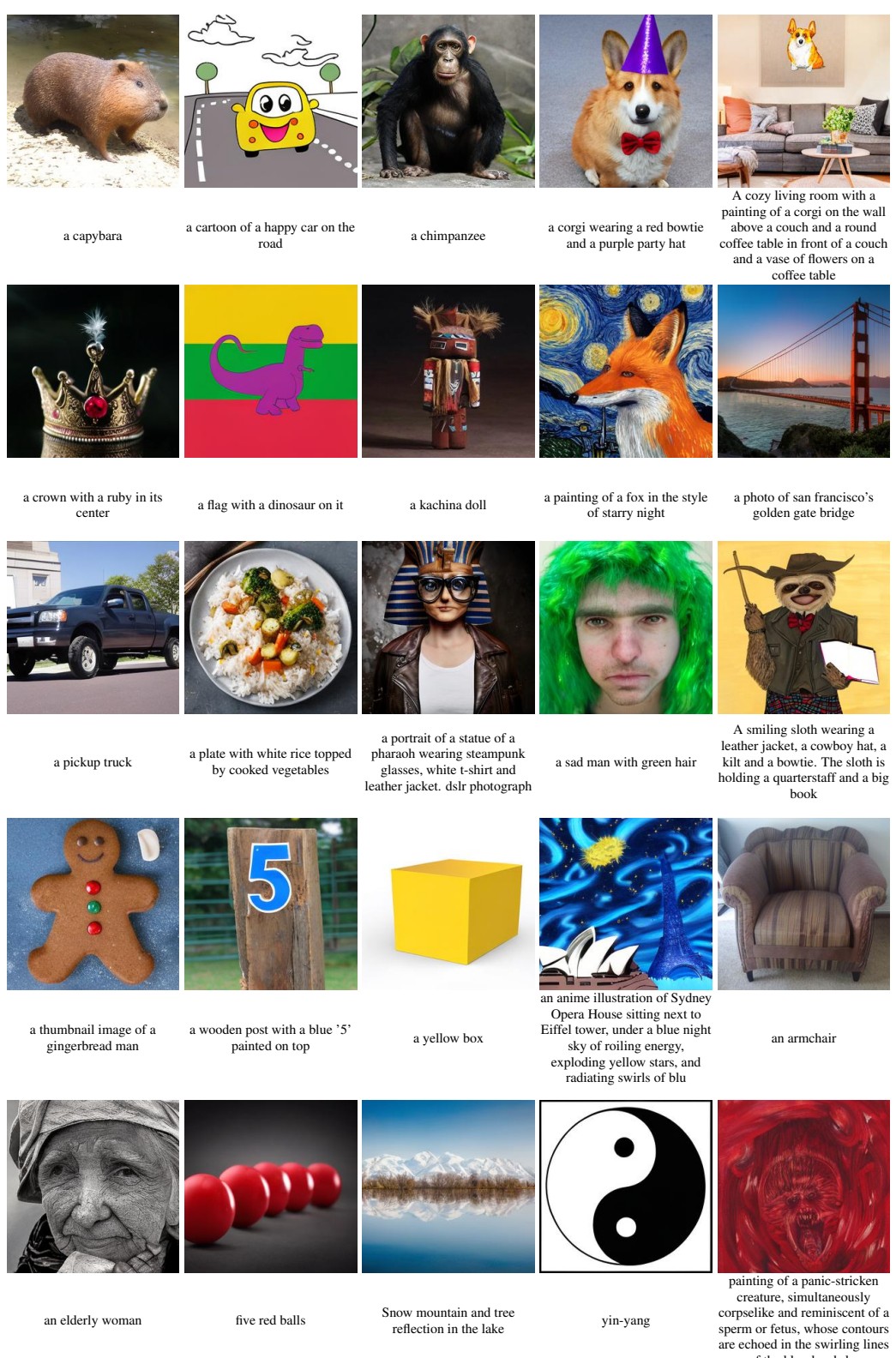

Figure 20: **Qualitative Results of Our Method on Text-to-Image Generation at 256×256 Resolution.** The input text prompts are shown below the images. Results are obtained from generative models trained for 200K steps.

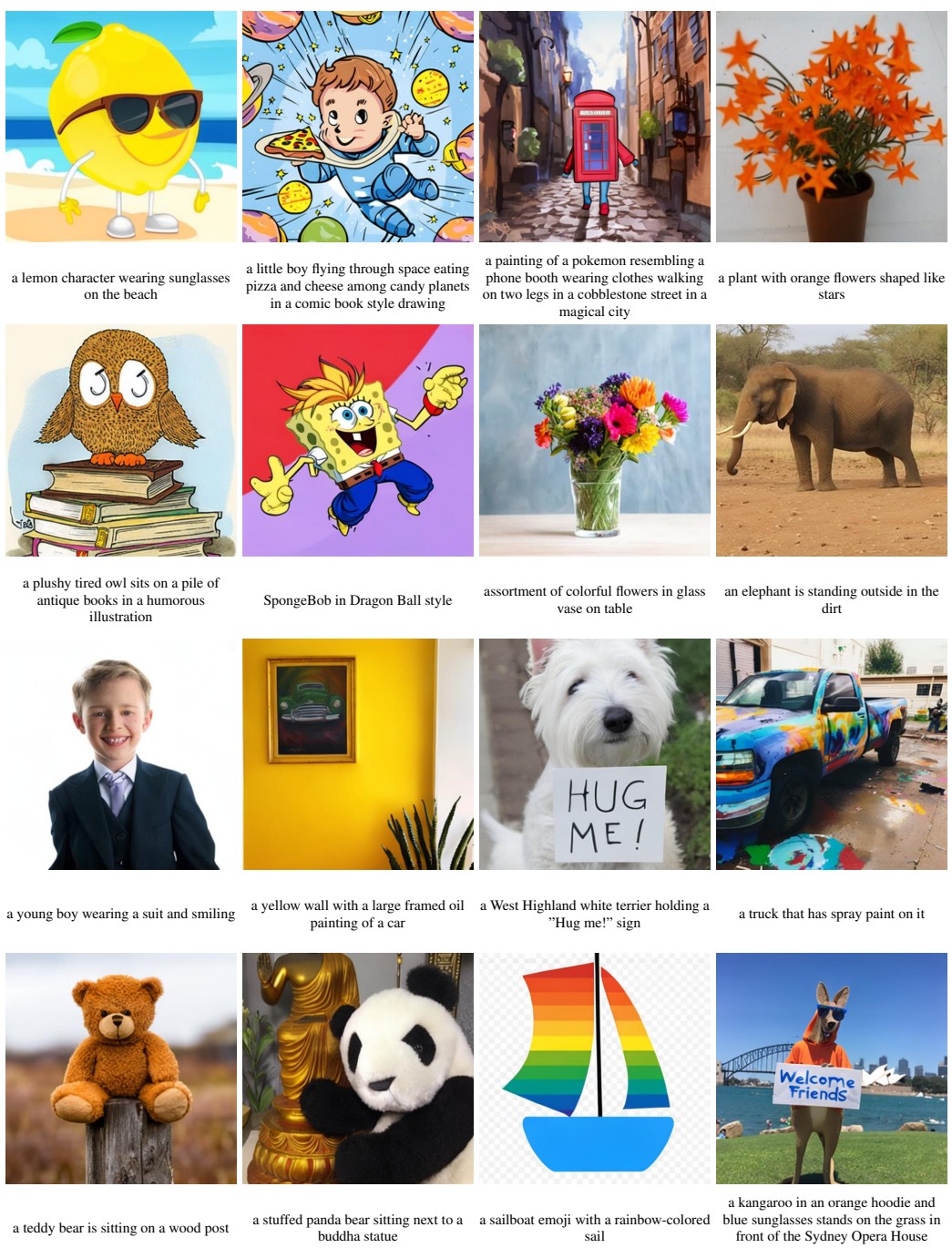

Figure 21: **Qualitative Results of Our Method on Text-to-Image Generation at 512×512 Resolution.** The input text prompts are shown below the images. Results are obtained from generative models trained for 290K steps.

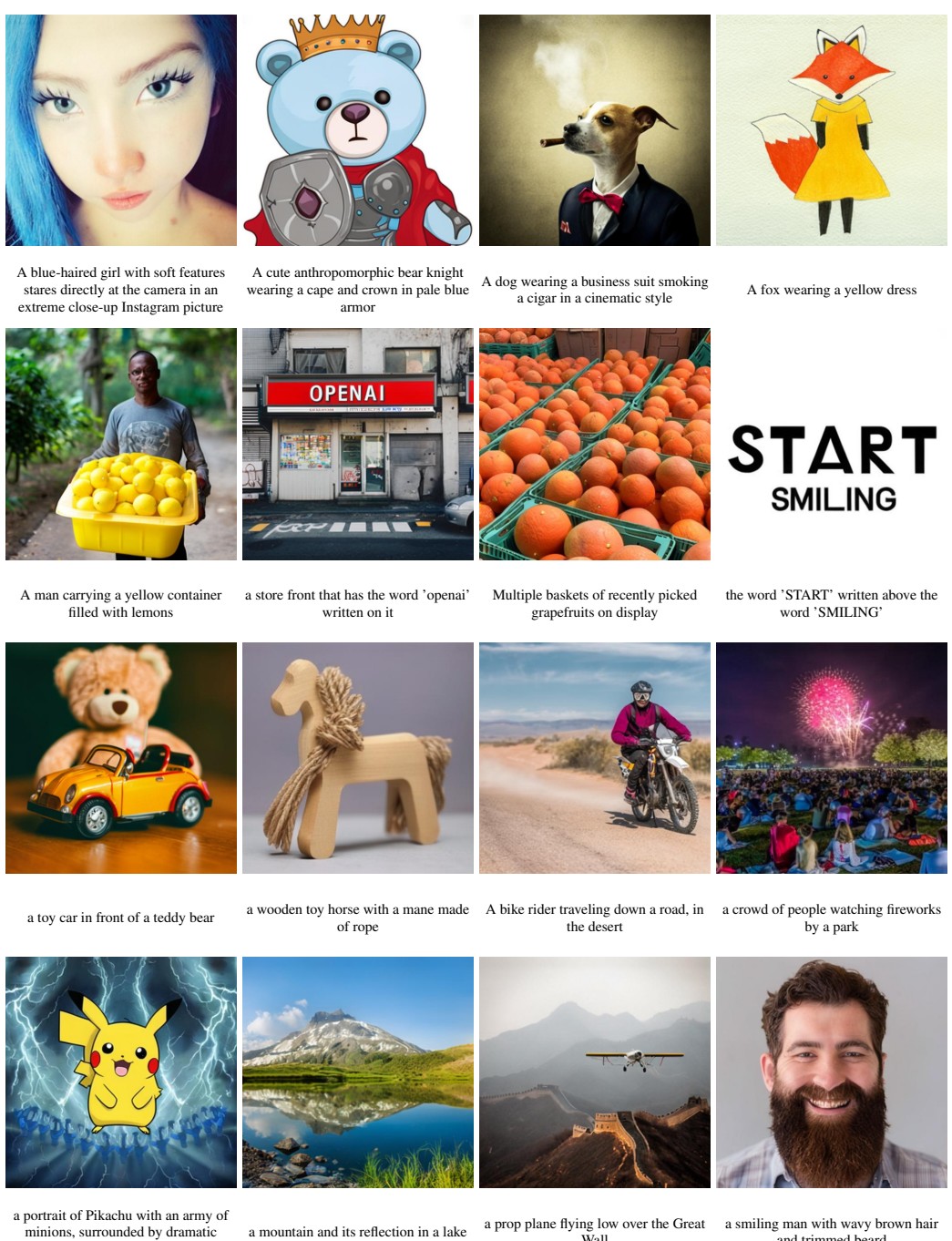

Figure 22: **Qualitative Results of Our Method on Text-to-Image Generation at 512×512 Resolution.** The input text prompts are shown below the images. Results are obtained from generative models trained for 290K steps.

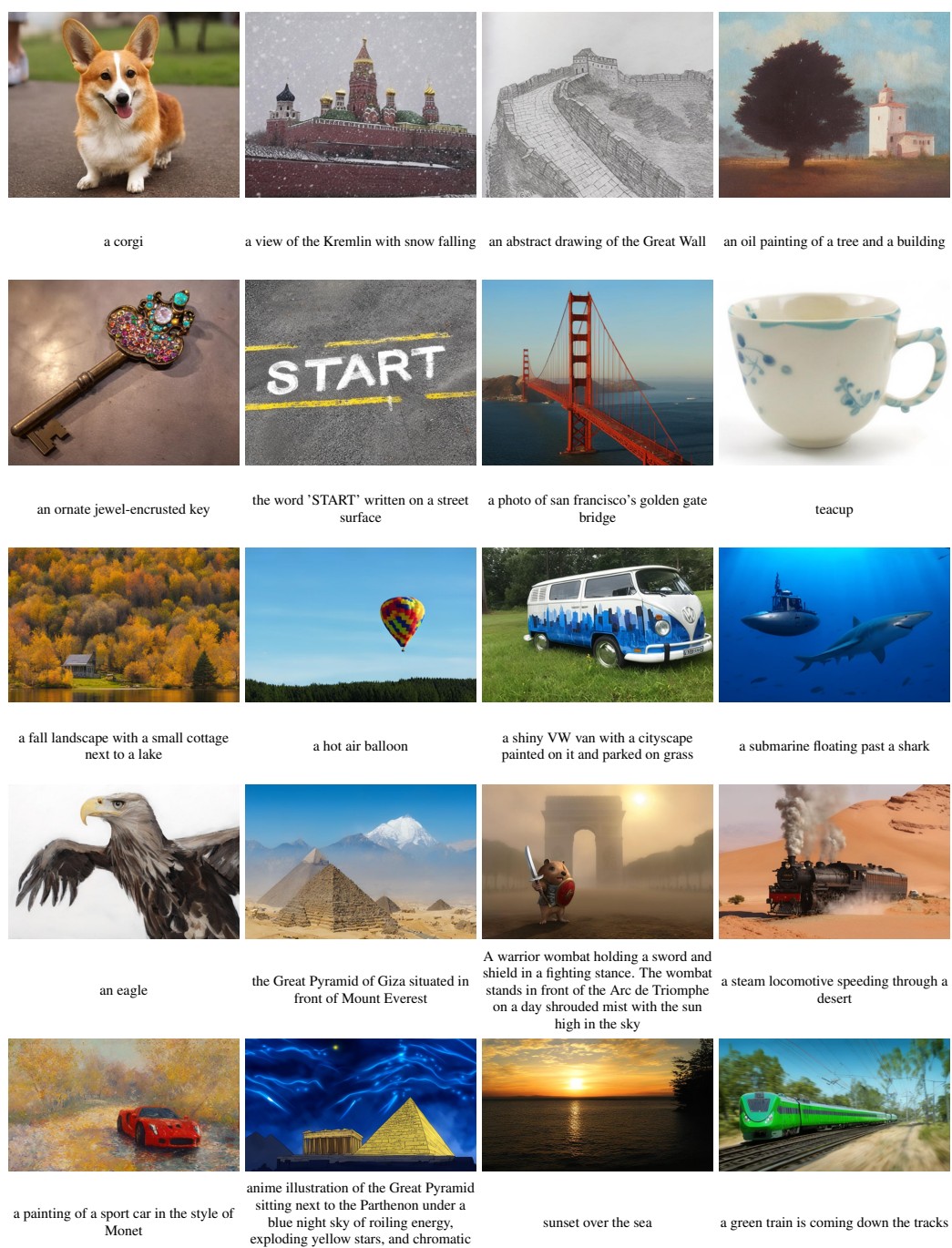

Figure 23: **Qualitative Results of Our Method on Text-to-Image Generation at Various Aspect Ratios with the Shortest Edge Equal to 512.** The input text prompts are shown below the images. Results are obtained from generative models trained for 290K steps. Rows 1 and 2 show 512 × 640 (3:4), row 3 and 4 show 512 × 768 (2:3), and row 5 shows 512 × 910 (9:16).

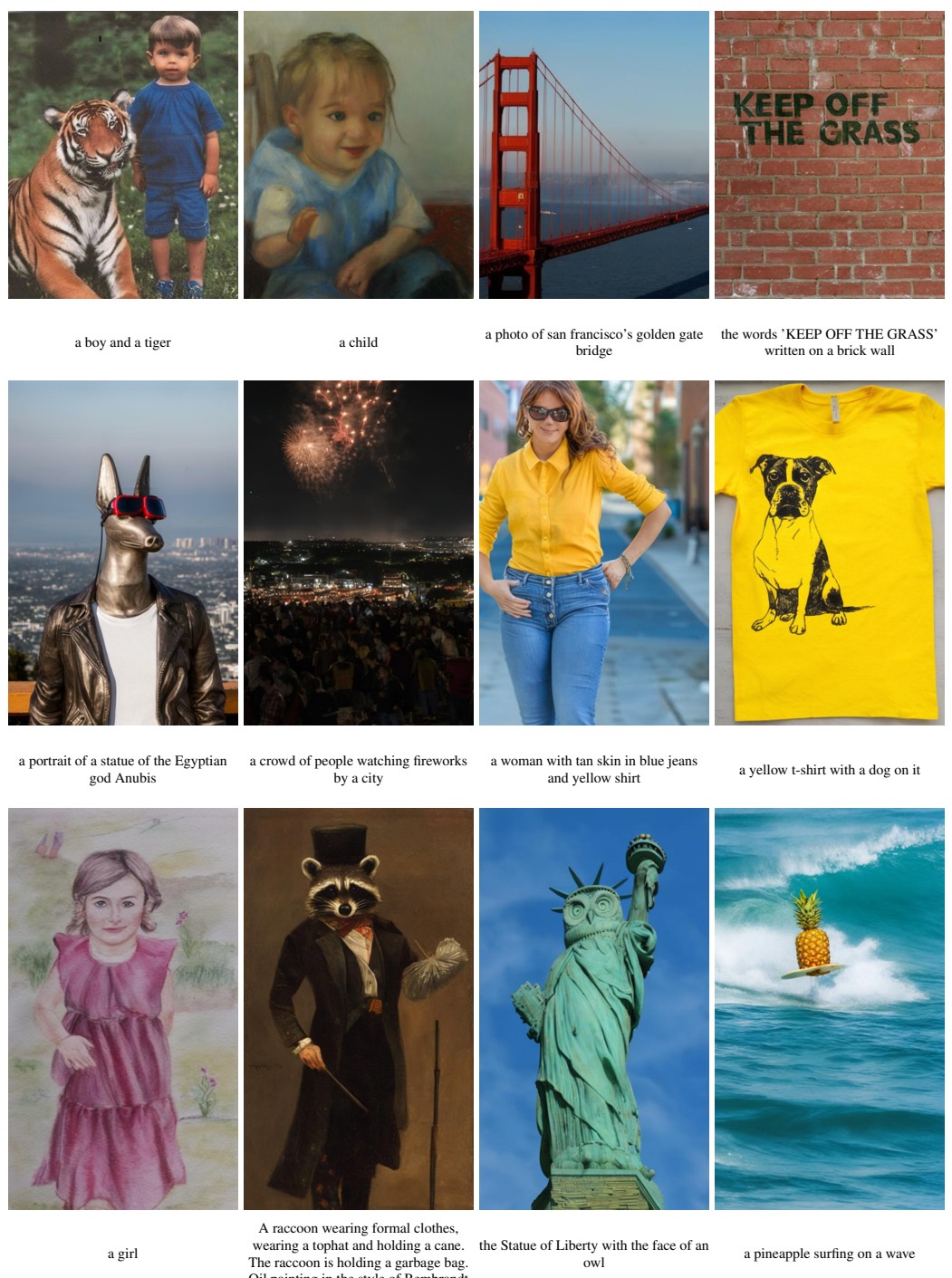

Figure 24: **Qualitative Results of Our Method on Text-to-Image Generation at Various Aspect Ratios with the Shortest Edge Equal to 512.** The input text prompts are shown below the images. Results are obtained from generative models trained for 290K steps. Rows 1 and 2 show $640 \times 512$ (4:3), rows 3 and 4 show $768 \times 512$ (3:2), and row 5 shows $910 \times 512$ (16:9).

