# OpenReview forum: "Aligning Visual Foundation Encoders to Tokenizers for Diffusion Models"
_ICLR.cc/2026/Conference — ICLR 2026 Poster_

### Official Review · Reviewer_LVhM · 2025-10-27

**Soundness:** 3
**Presentation:** 2
**Contribution:** 3
**Rating:** 4
**Confidence:** 4

**Summary:**

This paper proposes replacing the tokenizer VAE in LDM with pretrained visual encoders, specifically DINOv2, and designs a three-stage training framework. Extensive experiments demonstrate the effectiveness of the proposed method.

**Strengths:**

1. The proposed approach is simple yet effective which leverages a pretrained DINO model as the encoder and introduces a three-stage training scheme that balances semantic representation and reconstruction quality.
2. Comprehensive experiments verify the effectiveness of the method.

**Weaknesses:**

1. The motivation, *“Our intuition is that learning semantics is inherently more difficult than learning reconstruction”*,  requires further clarification and supporting evidence.

2. My main concern is that all training and evaluation are conducted on ImageNet without any OOD testing. It is possible that the low gFID results stem from overfitting rather than genuine improvements in generalization, since introducing semantic latent distributions may inherently simplify the latent structure compared to pixel-level ones [1,2].

3. Moreover, the authors use DINOv2 as a frozen encoder, which is pretrained on hundreds of millions of images[3]. This model may already include exposure to datasets similar to ImageNet, making it unclear whether the observed gFID improvement truly arises from the proposed architecture itself, or simply from the pretrained encoder’s prior knowledge of similar data.

4. Although the proposed three-stage training improves reconstruction performance as much as possible, the reconstruction FID still degrades (see Table 5), which limits the applicability of the method as a general-purpose encoder for tasks requiring accurate reconstruction, such as image editing and inpainting.

5. The code and evaluation scripts should be released to ensure reproducibility.

[1] Reconstruction vs. generation: Taming optimization dilemma in latent diffusion models

[2] Masked Autoencoders Are Effective Tokenizers for Diffusion Models

[3] DINOv2: Learning Robust Visual Features without Supervision

**Questions:**

See Weaknesses.

---

> ### Author Response · Authors · 2025-11-23
> **First-Round Rebuttal for Reviewer LVhM; Part 1**
>
> Thank you for your insightful feedback. We appreciate your acknowledgement of our method’s simplicity and effectiveness. Below, we address your concerns.
>
> ## Clarification and Supporting Evidence for the Motivation
>
> ***Question**: The motivation, “Our intuition is that learning semantics is inherently more difficult than learning reconstruction”, requires further clarification and supporting evidence.*
>
> Our intuition stems from a fundamental difference between the two objectives. Reconstruction primarily captures local pixel correlations and is directly supervised with dense signals, making it relatively easy to optimize. In contrast, semantic learning requires discovering global structures and invariances across diverse contexts, which is an inherently more difficult and data-hungry objective.
>
> This is reflected in the compute required in practice. Training strong semantic encoders is far more demanding: DINOv2 \[1\] requires **22,016 A100-hours**, while reconstruction-focused VAEs require less compute (PixArt-α VAE \[2\]: **1,600 V100-hours**; MaskBit \[3\]: **3,456 A100-hours**). Despite differences across architectures and datasets, the magnitude gap suggests that semantic learning is harder than reconstruction.
>
> \[1\] Oquab, Maxime, et al. “DINOv2: Learning Robust Visual Features without Supervision.” arXiv preprint arXiv:2304.07193 (2023).
> \[2\] Chen, Junsong, et al. “PixArt-α: Fast Training of Diffusion Transformer for Photorealistic Text-to-Image Synthesis.” ICLR, 2024\.
> \[3\] Weber, Mark, et al. “MaskBit: Embedding-Free Image Generation via Bit Tokens.” Transactions on Machine Learning Research, 2024\.
>
> ## OOD Testing
>
> For the rebuttal, we train 1B-parameter text-to-image (T2I) models on LAION-2B for 50K steps, comparing VA-VAE and our tokenizer under identical settings (the tokenizer is the only component that differs). Importantly, both tokenizers are trained *only* on ImageNet, ensuring that this experiment measures OOD generalization.
>
> **We add Tab. 14 and Tab. 15 in the updated paper**, which report quantitative results on *Parti Prompt*, *COCO Prompt 6K*, *HPSv2 Prompt*, and *GenEval*, for T2I models with and without classifier-free guidance (CFG). Across all reported metrics, our tokenizer consistently outperforms VA-VAE. In addition, **we include Fig. 18 in the updated paper for qualitative comparisons**, showing that our method produces images with higher visual quality and stronger prompt alignment than VA-VAE. Together, these results indicate that our alignment-based training strategy generalizes more effectively than the VA-VAE objective, which trains the encoder from scratch while encouraging its latent space to resemble DINOv2 features.
>
> **In addition, we compared against the non-semantic FLUX-VAE tokenizer. Please refer to Tab. 5, Tab. 11–13, and Fig. 5, Fig. 13-17.** Overall, our method achieves better quantitative and qualitative performance compared to FLUX-VAE.
>
> ## Improvement from Proposed Architecture or Prior Knowledge?
>
> **Question:** *Moreover, the authors use DINOv2 as a frozen encoder, which is pretrained on hundreds of millions of images. This model may already include exposure to datasets similar to ImageNet, making it unclear whether the observed gFID improvement truly arises from the proposed architecture itself, or simply from the pretrained encoder’s prior knowledge of similar data.*
>
> We clarify that the observed improvements do not simply arise from the pretrained DINOv2 encoder alone, but from our architectural design that effectively leverages this prior through the proposed alignment strategy. Below, we provide the justification.
>
> First, VA-VAE uses the same frozen DINOv2 encoder, yet performs consistently worse than ours, **including in the OOD T2I results shown in Tab. 14 and Tab. 15**. If the improvements stemmed merely from pretrained prior knowledge, VA-VAE should achieve comparable performance. The fact that it does not indicate that our alignment architecture, rather than the prior alone, is responsible for the gains.
>
> Second, our *Stage 1 only* variant in Tab. 1 relies even more directly on the DINOv2 prior, but still falls short in generation quality. This further demonstrates that pretrained priors alone are insufficient and that architectural design is essential.
>
> Conversely, the alignment would not be effective without a strong pretrained encoder. **Our method succeeds because it leverages both (1) the rich semantic priors from DINOv2 and (2) an alignment mechanism that transforms these priors into a latent space suitable for generative modeling.**

---

> > ### Author Response · Authors · 2025-11-23
> > **First-Round Rebuttal for Reviewer LVhM; Part 2**
> >
> > ## Degradation in rFID
> >
> > **Question:** *Although the proposed three-stage training improves reconstruction performance as much as possible, the reconstruction FID still degrades (see Table 5), which limits the applicability of the method as a general-purpose encoder for tasks requiring accurate reconstruction, such as image editing and inpainting.*
> >
> > We acknowledge that the current tokenizer does not yet reach the reconstruction fidelity of reconstruction-oriented VAEs. At the same time, our experiments show clear benefits of constructing a semantically structured latent space, and we view this work as a step toward tokenizers that simultaneously support strong semantics and high-quality reconstruction.
> >
> > Importantly, the current reconstruction level is not a fundamental limitation of our framework. Two extensions can further improve reconstruction fidelity:
> >
> > 1. **Increasing the latent channel dimensionality**, as enabled by RAE \[4\], which expands the model’s representational capacity and can consequently improve both reconstruction fidelity and semantic capacity. While RAE enables high-channel latent spaces, its training strategy freezes the encoder, which limits reconstruction quality (rFID 0.52 vs. our 0.26 under the same DINOv2 encoder). We can see a strong potential in combining RAE with our approach to build a tokenizer that achieves both high reconstruction fidelity and rich semantic capacity.
> > 2. **Replacing the decoder in Stage 3 with a larger and more expressive one**, which can better utilize the latent representation while keeping the encoded semantics fixed.
> >
> > These directions are complementary to our core contribution and offer promising avenues for future improvement.
> >
> > \[4\] Zheng, Boyang, et al. “Diffusion Transformers with Representation Autoencoders.” arXiv preprint arXiv:2510.11690 (2025).
> >
> > ## Code Release
> >
> > Yes, we will release the code and evaluation scripts upon acceptance.

---

### Official Review · Reviewer_GRL2 · 2025-11-01

**Soundness:** 3
**Presentation:** 3
**Contribution:** 2
**Rating:** 4
**Confidence:** 4

**Summary:**

- Generation models (e.g., Diffusion models, flow matching model) employ an autoencoder to reduce the dimension of the image dataset for computational efficiency and generation ability.
- The authors claim that the latent space of the encoder should be semantically meaningful.
- By introducing DINO-v2 initialized encoder and semantic preservation regularization, they achieve faster convergence while training and better fid with the same NFE while inference.

**Strengths:**

- The author first proposed to directly employ the pretrained vision foundation model, followed by an adaptor as an encoder for training a generative model.
- The proposed architecture is simple but effective.
- The experiments in Table 1 support the effects of each configuration. Specifically, in the Semantic preservation loss table, there is a trade-off relationship between the reconstruction and generation performance
- Compared to the VA-VAE, the proposed method achieves better gFID along various sampling steps and CFG scale. In addition, the proposed latent space makes the generative models converge faster than the latent space of VA-VAE
- Compared to FLUX VAE, the proposed methods achieve better generation performance qualitatively and quantitatively.

**Weaknesses:**

- The overall training strategy seems heuristic, falling short of scalability. Combining multiple stages into a simple pipeline or providing principled methods to measure when to stop each stage would be helpful for applicability.
- Why DINO-v2 works better than other foundation models? An analysis or theoretical explanation would be helpful
- Lack of analysis of the semantics of the latent space.
- [REPA-E] also incorporates the vision foundation model for training VAE. What is the advantage of the proposed method over the [REPA-E]?

[REPA-E]: REPA-E: Unlocking VAE for End-to-End Tuning with Latent Diffusion Transformers

**Questions:**

- What is the author's thought about the “generative-friendly latent space”? According to the claim of the paper, it seems that preservation of the latent space of DINOv2 is helpful for training generative models. If so, would it be an optimal and promising direction of constructing latent space to only get closer to the DINO space? Are there any other components that should be considered?
- What  ‘linear probing’ exactly did you use for evaluation, and why? Do all the measurements for linear probing give the same correlation and conclusion?
- The vision foundation model could be huge and computationally expensive. Can you compare the size of the proposed tokneizer with the others?

---

> ### Author Response · Authors · 2025-11-23
> **First-Round Rebuttal for Reviewer GRL2; Part 1**
>
> Thank you for your insightful feedback. We appreciate your acknowledgement of our method’s simplicity and its proven effectiveness. Below, we respond to your concerns.
>
> ## Training Strategy
>
> Our principle to measure when to stop follows directly from our objective: we aim to construct a tokenizer with both strong semantic capacity and high reconstruction fidelity. Because our method starts from a pretrained encoder and includes mechanisms to preserve semantics, the semantic capacity remains overall stable throughout training (see Fig. 3, right). As a result, the quantity that continues to improve over the stages is the reconstruction fidelity. This leads to a principled and objective stopping criterion: **for each stage, we monitor reconstruction quality and terminate the stage once its improvement saturates.**
>
> It is also important to note that determining a stopping point for vanilla VAE and VA-VAE is challenging, even though they use only a single-stage pipeline. Since they are trained from scratch, their latent spaces may not yet be generative-friendly or semantically meaningful at the point when reconstruction appears good, making early stopping unreliable. In contrast, our three-stage alignment method decouples semantic learning from reconstruction learning, leading to a more controlled and interpretable optimization process.
>
> ## Why DINOv2 Works Better than Other Foundation Models
>
> We think this is because DINOv2 captures a more effective balance between global semantic structure and local spatial detail than other foundation models. This is supported by empirical evidence: DINOv2 \[1\] outperforms MAE \[2\] and SigLIP2 \[3\] on dense prediction tasks such as segmentation and depth, as reported in their papers.
>
> Mechanistically, DINOv2’s objective encourages global semantic grouping while maintaining local spatial consistency. In contrast, MAE uses a reconstruction loss, which may encourage the model to focus more on local details, and SigLIP2 focuses on global image-text alignment, which can down-weight fine-grained spatial information. Consequently, DINOv2 strikes a more favorable balance between global semantics and local spatial coherence. A tokenizer built on DINOv2 inherits these properties, yielding latent representations that are both semantically aligned and spatially coherent, ultimately improving generation quality.
>
> \[1\] Oquab, Maxime, et al. “DINOv2: Learning Robust Visual Features without Supervision.” arXiv preprint arXiv:2304.07193 (2023).
> \[2\] He, Kaiming, et al. “Masked Autoencoders Are Scalable Vision Learners.” CVPR 2022\.
> \[3\] Tschannen, Michael, et al. “SigLIP 2: Multilingual Vision-Language Encoders with Improved Semantic Understanding, Localization, and Dense Features.” arXiv preprint arXiv:2502.14786 (2025).
>
> ## Analysis of the Semantics of the Latent Space
>
> We provided the linear probing accuracy in Tab. 1, Tab. 3 and Fig. 3 to evaluate the semantic of the latent space. The linear probing accuracy is obtained by training a linear layer on top of the latent space for ImageNet image classification task. This is a standard and widely used measure of semantic richness in the visual representation learning community.
>
> For the rebuttal, **we add PCA visualizations of the latent space in Fig. 7 of the updated paper**. The figure compares the latent spaces produced by three tokenizers (Vanilla VAE, VA-VAE, and Ours) with the output of the pretrained DINOv2 encoder as a semantic reference. Vanilla VAE tends to produce latents that are either overly smooth or overly sharp, while VA-VAE exhibits oversmoothing. In contrast, our latent space most closely resembles the structure of DINOv2 features, reflecting its stronger semantic structure.
>
> Besides, **we also include a quantitative analysis of the latent space in Tab. 16 of the updated paper.** We evaluate several distributional metrics such as CKNNA \[4\], Total Variation, and Gini Coefficient. While our latent space aligns more closely with the DINOv2 space than other tokenizers in metrics like CKNNA and Total Variation, VA-VAE appears closest in terms of metrics like Gini Coefficient. We note, however, that each of these metrics captures only one aspect of the latent distribution, and none of them individually correlates perfectly with semantic quality. This is why we relied primarily on linear probing to evaluate the semantic of the latent space.
>
> \[4\] Huh, Minyoung, et al. “The Platonic Representation Hypothesis.” ICML, 2024\.

---

> > ### Author Response · Authors · 2025-11-23
> > **First-Round Rebuttal for Reviewer GRL2; Part 2**
> >
> > ## REPA-E
> >
> > While both REPA-E \[5\] and our method leverage a vision foundation model, it is not straightforward to claim an overall advantage of one approach over the other because the two methods follow different setups and can in fact be combined. Below are the explanations.
> >
> > REPA-E trains the tokenizer and the diffusion model jointly in an end-to-end manner. Even under this end-to-end regime, the authors show that initializing the tokenizer with an existing tokenizer works better than random initialization; specifically, their best model (E2E-VAE) initializes the tokenizer from VA-VAE. In contrast, our method focuses solely on training the tokenizer itself. **This makes our tokenizer a drop-in replacement for the VA-VAE component inside REPA-E: since REPA-E already benefits from initializing its tokenizer with VA-VAE, it could similarly initialize with our tokenizer as a potentially stronger starting point for its end-to-end training.**
> >
> > \[5\] Leng, Xingjian, et al. “REPA-E: Unlocking VAE for End-to-End Tuning with Latent Diffusion Transformers.” arXiv preprint arXiv:2504.10483 (2025).
> >
> > ## Thoughts about Generative-Friendly Latent Space
> >
> > ***Questions**: What is the author's thought about the “generative-friendly latent space”?*
> >
> > A generative-friendly latent space should be semantically rich while still retaining perceptual, low-level details. Semantics help the diffusion model understand global structure, while perceptual cues are essential for faithful reconstruction and high-frequency details.
> >
> > ***Questions**: According to the claim of the paper, it seems that preservation of the latent space of DINOv2 is helpful for training generative models. If so, would it be an optimal and promising direction of constructing latent space to only get closer to the DINO space? Are there any other components that should be considered?*
> >
> > Regarding whether the optimal latent space should simply match the DINOv2 space: **not necessarily**. While DINOv2 provides strong semantic structure, it intentionally discards many low-level visual details and is not designed for reconstruction. Therefore, the goal is not to reproduce DINOv2 exactly, but to approach it in a balanced way that preserves semantics without sacrificing perceptual information.
> >
> > This is reflected in our ablation results in Tab. 1\. The *Stage 1 only* variant produces latents that are closest to the DINOv2 space, yet its generative performance is worse than *Stage 1 \+ Stage* 2, which maintains more perceptual detail while being slightly farther from DINOv2. This suggests that the optimal latent space lies between pure DINOv2 semantics and pure reconstruction fidelity, rather than at either extreme.
> >
> > ## Linear Probing
> >
> > ***Question**: What ‘linear probing’ exactly did you use for evaluation, and why?*
> >
> > For linear probing, we follow the standard ImageNet protocol widely used to assess the semantic quality of latent representations. Specifically, we freeze both the VAE encoder and the adapter, and train a single linear classifier on top of the adapter’s output features to predict ImageNet-1K classes. We use linear probing because it directly measures how much semantic information is linearly accessible in the latent space, making it a widely adopted proxy for semantic quality. The classifier is trained for 3 epochs using Adam optimizer with a learning rate of 0.001 and a batch size of 256 on the ImageNet-1K training set, and evaluated on the ImageNet validation set. **These details have been added to the updated paper.**
> >
> > ***Question**: Do all the measurements for linear probing give the same correlation and conclusion?*
> >
> > Yes. We ran linear probing five times for both our tokenizer and VA-VAE using different random seeds. The results are highly consistent, with standard deviations of 0.24% for our tokenizer and 0.17% for VA-VAE, confirming that all measurements lead to the same conclusions.
> >
> > ## Comparison of The Size of the Proposed Tokenizer with The Others
> >
> > We provide a table below to compare the size of the proposed tokenizer with our baselines. More detailed comparisons of training and inference cost, such as training GPU hours, encoder latency, and GPU memory consumption, are provided in *[A. Response to Training and Inference Cost](https://openreview.net/forum?id=ajnBafpqmE\&noteId=PO4t1oxS5W).*
> >
> > We acknowledge that our encoder has more parameters and higher latency than the other tokenizers. However, as shown in the detailed analysis, it requires fewer GPU hours to train and consumes less GPU memory at larger batch sizes and higher resolutions during inference. Besides, our experiments show that our tokenizer provides substantial benefits for diffusion model training.
> >
> > **Table: Comparison of the Size of Tokenizers.**
> >
> > | Tokenizer | Enc. \#Params | Dec. \#Params |
> > | :---: | :---: | :---: |
> > | **Vanilla VAE** | 28.41 M | 41.42 M |
> > | **VA-VAE** | 28.41 M | 41.42 M |
> > | **Ours** | 304.4 M | 41.42 M |

---

### Official Review · Reviewer_svcV · 2025-11-03

**Soundness:** 3
**Presentation:** 3
**Contribution:** 2
**Rating:** 4
**Confidence:** 4

**Summary:**

The authors propose a three stage framework to transform a pre-trained semantic encoder into an image tokenizer. In the first stage, they employ an adaptor that maps features from the frozen encoder into a lower-dimensional latent space (32 dimensions in their experiments). The adaptor and decoder are jointly trained using a reconstruction loss. In the second stage, the encoder is fine-tuned to improve reconstruction quality. However, since this process can degrade semantic capacity, an L2 regularization term is applied on the low-dimensional latent to keep the updated encoder close to its previous representation. Experiments conducted mainly on ImageNet 256 show a 0.46 improvement in FID when using the LightningDiT framework.

**Strengths:**

The presentation, though sometimes not self-contained, is overall clear.

The empirical performance, evaluated on ImageNet, shows 3.4% improvements on linear probing, on ImageNet 256.

**Weaknesses:**

1. It's not clear what the overall computational cost is for all stages, when compared to other methods.

2. I am not sure how sensitive the dimensionality of the adaptor is. If we want to generalize to ImageNet 512, does it mean that we only need to double the latent?

3. The use of an L2 loss to preserve semantic consistency may not be robust, especially when the latent space lacks proper normalization.

4. While semantic preservation appears improved, the drop in linear probing accuracy during training raises concerns about the trade-off between reconstruction fidelity and semantic capacity.

5. The decrease with rFID shows that the constraint on the adaptor and the stage-wise framework might impact the reconstruction accuracy.

**Questions:**

What's the performance before CFG?

---

> ### Author Response · Authors · 2025-11-23
> **First-Round Rebuttal for Reviewer svcV; Part 1**
>
> Thank you for your insightful feedback and comments. We appreciate your acknowledgement of our method’s effectiveness. Below, we respond to your concerns.
>
> ## Computational Cost
>
> We would like to first emphasize that our method consists of three stages during training. Once the tokenizer is fully trained, it performs encoding and decoding with a single forward pass, the same as other tokenizers.
>
> For a detailed comparison of the training cost across the three stages, as well as the inference cost, please refer to *[A. Response to Training and Inference Cost](https://openreview.net/forum?id=ajnBafpqmE\&noteId=PO4t1oxS5W).*
>
> ## Dimensionality of the Adapter
>
> The adapter compresses the high-dimensional encoder feature (1024 channels) into a low-dimensional latent space (we use 32 channels by default). We adopt 32 channels because, at the time of writing, diffusion models were known to perform reliably only with low-dimensional latents.
>
> **This 32-channel configuration is robust and resolution-agnostic.** If we want to generalize to ImageNet 512, we can keep the latent dimension at 32; only the spatial size of the latents increases because of the larger input resolution under the same downsampling ratio. The table below illustrates this difference.
>
> The robustness of 32-channel latents at higher resolution is further validated by our 512-resolution text-to-image results in Fig. 6 of the paper, where we set the latent channel dimension to 32\.
>
> ####
>
> **Table: Illustration of Latent Shape Scaling with Resolution.** We set the channel dimension to 32\.
>
> | Input Image Shape | Downsampling Ratio | Output Latent Shape |
> | :---: | :---: | :---: |
> | 256 × 256 x 3 | 16 | 16 x 16 x 32 |
> | 512 × 512 x 3 | 16 | 32 x 32 x 32 |
>
> ## Loss with Normalization
>
> Thank you for the suggestion, and this is a good point. To address the potential sensitivity of L2 loss to scale, we added a variant that replaces the L2 loss with a cosine similarity loss, which applies the semantic constraint with explicit normalization. As shown in the table below, the cosine-loss variant achieves performance comparable to our original design. **These results have been added to Tab. 1 of the revised paper.**
>
> This experiment suggests that, at least in our current setup, both losses lead to similar behavior, indicating that the performance differences are not dominated by the specific choice of L2 versus cosine loss. We will explore more principled semantic-preservation objectives in future work.
>
> **Table: Effect of Loss Normalization.** Evaluated on ImageNet 256×256 at 80K training steps with 30 sampling steps, using the CFG scale that yields the lowest generation FID (gFID).
>
> | Configuration | rFID ↓ | PSNR ↑ | LPIPS ↓ | L. P. Acc. ↑ | gFID ↓ | IS ↑ | Prec ↑ | Recall ↑ |
> | :---: | :---: | :---: | :---: | :---: | :---: | :---: | :---: | :---: |
> | Cosine Loss | 0.37 | 25.23 | 0.129 | 37.99% | 2.23 | 248.6 | 0.819 | 0.585 |
> | L2 Loss | 0.36 | 25.62 | 0.121 | 35.09% | 2.19 | 248.6 | 0.811 | 0.591 |
>
> ## Trade-Off Between Semantics and Reconstruction
>
> We acknowledge that, in practice, there is an inherent trade-off between reconstruction fidelity and semantic capacity for every tokenizer. Our experiments show that a more semantic latent space brings clear benefits, such as improved training convergence and reduced CFG scale. For this reason, we believe that vanilla VAE that focuses solely on reconstruction quality is not an optimal design choice. Although this trade-off cannot be eliminated, our goal is to move the Pareto frontier by constructing a latent space that better balances semantic structure and reconstruction fidelity.
>
> While this trade-off remains, a promising future direction is to increase the latent channel dimensionality, thereby expanding the model’s capacity and shifting the Pareto frontier upward to improve both reconstruction fidelity and semantic richness. Importantly, training diffusion models on higher-channel latent spaces becomes feasible, as demonstrated by the concurrent work RAE \[1\]. While RAE enables the use of high-channel latent spaces, its specific training strategy freezes the encoder, which limits reconstruction quality (rFID 0.52 vs. our 0.26 using the same DINOv2 encoder). This suggests that combining our method with the high-channel latent design enabled by RAE can help build a tokenizer with both strong semantic capacity and reconstruction fidelity.
>
> \[1\] Zheng, Boyang, et al. “Diffusion Transformers with Representation Autoencoders.” arXiv preprint arXiv:2510.11690 (2025).

---

> ### Author Response · Authors · 2025-11-23
> **First-Round Rebuttal for Reviewer svcV; Part 2**
>
> ## The Adapter and the Stage-Wise Framework Impact Reconstruction
>
> The drop in rFID is expected. As the adapter compresses the encoder features into a low-channel latent space, some fine-grained information is inevitably lost, which affects reconstruction accuracy. In addition, our stage-wise framework explicitly preserves the semantic structure of the latent space, which can impact reconstruction fidelity.
>
> As discussed in the previous response, increasing the latent channel dimensionality is a fundamental way to expand the model’s capacity, effectively making the cake bigger so that both reconstruction and semantics can improve together. A complementary direction is to replace the Stage 3 decoder with a more powerful one. Since the encoder (and thus the semantic latent space) is fixed at this stage, a stronger decoder can better recover high-quality reconstructions from the same latent codes.
>
> ## Performance before CFG
>
> We have provided ImageNet evaluations of our model without classifier-free guidance (CFG) in Tab. 3, Tab. 4, and Fig. 4\. For the rebuttal, we included further comparisons of text-to-image models without CFG. **Please refer to the newly added Tab. 12–15 in the updated paper for non-CFG quantitative comparisons against FLUX-VAE and VA-VAE. In addition, we include Fig. 17 to provide qualitative comparisons under the non-CFG setting with FLUX-VAE. In summary, our method consistently outperforms the baselines both quantitatively and qualitatively.**

---

### Official Review · Reviewer_XEQ8 · 2025-11-04

**Soundness:** 3
**Presentation:** 3
**Contribution:** 3
**Rating:** 6
**Confidence:** 4

**Summary:**

This paper introduces a new way of image tokenizer training for image generation. It targets the weak semantics of the latent space of traditional VAE tokenizers, which can be caused by the reconstruction-only training supervision. This method builds an image tokenizer based on pre-trained discriminative vision foundation models such as DINO. It introduces a progressive three-stage training method that trains an adaptor and a decoder to reconstruct the images, while maintaining the semantically rich latent space. Experiments on ImageNet and text-to-image generation prove the effectiveness. Extensive ablations justify the design of each training stage.

**Strengths:**

- This paper targets an important problem of semantic-rich image tokenizer traning. The solution shown is simple and effective. The idea is well presented and illustrated.

- The extensive ablation and detailed reconstruction / semantic capacity tracking along the training process are very helpful and insightful.

- The final quantivative evaluation results are strong.

**Weaknesses:**

- Not technically a weakness, but the final linear probing performance is only ~35%, dropping by a lot compared to the original DINO latent space. What could be the potential ways of further bridging the gap?

- In the stage 2 training, applying the regularization to the output of $E_p$ instead of $A$ seems to be a more intuitive way and can better preserve the semantics. Could the authors provide some insights on why that does not work well?

- In summary, my main concern is that, although the idea is motivated by leveraging a semantically rich encoder to obtain a more diffusable latent space, the substantial linear-probing performance drop and the observed trade-offs between reconstruction fidelity and semantic preservation make it unclear whether the improved results stem from a genuinely richer semantic space or merely from initializing the tokenizer with a well-pretrained encoder that can also be some other models.

**Questions:**

- I didn't find detailed comparisons for the costs. Will using the pretrained vision encoder introduce increased memory consumption and encoding latency?

- Could the authors provide GenEval score comparisons in Table 5? It will provide more comprehensive evaluations on alignment and better justify the core contribution of this paper.

- Figure 4 right. The baseline VA-VAE fails to train in full training epochs in both setting due to numerical issues. Would it be possible to add another baseline for full-scale comparisons? This is optional as it might require prohibitive time during the rebuttal period.

---

> ### Author Response · Authors · 2025-11-23
> **First-Round Rebuttal for Reviewer XEQ8; Part 1**
>
> Thank you very much for your thoughtful feedback. We are greatly encouraged by your acknowledgment of our method’s simplicity and effectiveness, the clarity of our paper, and the depth of our experimental analysis. Below, we address your concerns.
>
> ## Drop of Linear Probing Accuracy
>
> We would like to first point out that the original DINOv2 latent space has a channel dimension of 1024, whereas our latent space is reduced to 32 channels after the adapter’s compression. This reduction naturally lowers linear probing accuracy. We chose a smaller channel dimension because, at the time of writing, diffusion models were known to perform reliably only with low-dimensional latents. Since our primary focus is improving generative performance, we adopted this smaller latent dimensionality. We have updated the right plot in Fig. 3 to clearly reflect the difference in channel dimensions.
>
> A potential way to bridge this gap is to increase the channel dimension of the latent, which is now made feasible by RAE \[1\]. RAE shows that diffusion models can be trained directly on high-dimensional latent spaces, overcoming the historical limitation that diffusion models were reliable only with low-dimensional latents. This enlarged latent space provides higher representational capacity, which can simultaneously improve reconstruction fidelity and semantic capacity.
>
> We see strong potential in combining RAE with our approach. While RAE enables the use of high-channel latent spaces, its specific training strategy freezes the encoder, which limits reconstruction quality (rFID 0.52 vs. our 0.26 using the same DINOv2 encoder). In contrast, our semantic-preserving encoder fine-tuning maintains semantic structure and yields better reconstruction. This complementary behavior suggests that combining RAE’s high-dimensional latent design with our alignment-based training strategy could lead to a tokenizer that achieves both strong reconstruction fidelity and high semantic capacity.
>
> \[1\] Zheng, Boyang, et al. “Diffusion Transformers with Representation Autoencoders.” arXiv preprint arXiv:2510.11690 (2025).
>
> ## Not Regularizing the Output of the Encoder $E\_p$
>
> Our insight is as follows. In Stage 1, the encoder $E\_p$ is frozen while the adapter $A$ is trainable. Because $A$ is a shallow MLP and its input (the output of the frozen $E\_p$) is fixed, $A$ cannot drift far and naturally preserves semantics in the latent space. In Stage 2, however, both $E\_p$ and $A$ are trainable. As $E\_p$ updates, the input to $A$ keeps changing, making the training problem harder than in Stage 1\. This constantly shifting input distribution to $A$ can cause the latent space produced by $A$ to drift away from semantically meaningful representations. Therefore, regularizing only the output of $E\_p$ in this stage is insufficient because it does not constrain how $A$ transforms the evolving $E\_p$ outputs. To ensure the final latent space remains semantically rich, we apply the regularization on the output of $A$ in Stage 2, encouraging it to stay close to the output of $A$ in Stage 1, which effectively constrains both $E\_p$ and $A$.
>
> Our ablation study verified our insight. In Tab. 1 of the paper, we provided the quantitative evaluation of applying the regularization to the output of $E\_p$, referred to as the *Applied Pre-Adapter* variant. Compared to our adopted design (regularizing the output of adapter $A$), this variant yields worse linear probing accuracy (semantics) and worse generation performance.
>
> ## Improved Results from Semantic Latent Space or Initialization of Pretrained Models?
>
> Our improvement stems from constructing a semantically-rich latent space, rather than merely initializing from well-trained encoders. Below are the key justifications.
>
> In Tab. 1 of the paper, the variant *Weight \= 0* is initialized from the same pretrained encoder, but it removes our semantic preservation loss. Even with this initialization, its linear probing accuracy drops to 9.50%, indicating that the resulting latent space is not semantically rich. Consequently, this variant exhibits worse generation quality compared to our adopted design, which is initialized identically but with a semantically richer latent space (35.09% linear probing accuracy). This demonstrates that our improvement comes from semantically rich latent space, rather than merely relying on pretrained initialization.
>
> In summary, pretrained initialization is helpful but not sufficient: the key improvement comes from our three-stage alignment that explicitly shapes a semantic latent space. The initialization is flexible; other strong encoders, such as SigLIP 2 \[2\], can be used as shown in Tab. 2, and the semantic benefit persists.
>
> \[2\] Tschannen, Michael, et al. “SigLIP 2: Multilingual Vision-Language Encoders with Improved Semantic Understanding, Localization, and Dense Features.” arXiv preprint arXiv:2502.14786 (2025).

---

> > ### Author Response · Authors · 2025-11-23
> > **First-Round Rebuttal for Reviewer XEQ8; Part 2**
> >
> > ## Cost Comparison
> >
> > Please refer to *[A. Response to Training and Inference Cost](https://openreview.net/forum?id=ajnBafpqmE&noteId=PO4t1oxS5W)*.
> >
> > ## GenEval
> >
> > Thanks for the suggestions. In the updated paper, **we add GenEval comparisons with FLUX VAE (Tab. 13\) and VA-VAE (Tab. 15).** Our method consistently outperforms both baselines across all metrics.
> >
> > ## Fig. 4 Right
> >
> > We believe this is a misunderstanding. The two failures in full training are not both VA-VAE; they correspond to VA-VAE and our method *without* QKNorm. Our method achieves better gFID than VA-VAE in both the without QKNorm setting (prior to the loss spike) and the with QKNorm setting. Note that we used the official VA-VAE checkpoints released by the authors.

---

### Author Response · Authors · 2025-11-23
**To All Reviewers**

We thank all reviewers for their thoughtful and constructive feedback. In response, we have revised the paper and added new experiments and analyses. All newly added content is highlighted in blue. Below, we summarize the key updates:

1. Training and inference cost comparison with other tokenizers (Tab. 6, Tab. 7, and Tab. 8).
2. Comparison of text-to-image generation on GenEval (Tab. 13 and Tab. 15).
3. Comparison of text-to-image generation without CFG (Tab. 12-15, Fig. 17).
4. Comparison of text-to-image generation with VA-VAE (Tab. 14, Tab 15, Fig. 18).
5. Qualitative and quantitative analysis of latent space (Fig. 7 and Tab. 16).

*For general comments, we provide them in a separate section. Reviewer-specific responses are given as replies to each individual review.*

---

### Author Response · Authors · 2025-11-23
**A. Response to Training and Inference Cost**

For the rebuttal, **we add the cost comparison of both training and inference in Tab. 6, Tab. 7, and Tab. 8 of the updated paper.**

To compare training cost, we show a simplified version of Tab. 6 below for convenience. Please check out the complete version with trainable and frozen parameters in the revised paper. Our main observations are as follows:

1. **Total GPU memory**. The vanilla VAE consumes the least memory because it does not load any pretrained encoder. Our Stage 1 and Stage 3 use less memory than VA-VAE since our encoder is frozen during these stages, whereas VA-VAE \[1\] loads CNN encoder (trainable) \+ DINOv2 encoder (frozen). Our Stage 2 consumes the most memory because it fine-tunes the DINOv2 encoder.
2. **Time per training step**. The pattern mirrors memory usage for the same reason: VA-VAE incurs higher cost than our Stage 1 and 3, and our Stage 2 incurs the highest cost when the encoder is trainable.
3. **Total A100 GPU hours**. The cumulative **GPU training hours of our method (107.2 \+ 377.4 \+ 91.55 \= 576.15) are lower than those of both Vanilla VAE and VA-VAE**, demonstrating the overall training efficiency of our pipeline.

**Table: Training Cost Comparison.** All measurements are obtained at an input resolution of 256$\\times$256 on 8 NVIDIA A100 GPUs (batch size 8 per GPU). Values marked with $\*$ denote GPU-hour estimates computed from the training epochs reported in the official implementation of VA-VAE \[1\].

| Tokenizer | Enc. \#Param  | Total GPU Memory | Time Per Training Step | Total A100 GPU Hours |
| :---: | :---: | :---: | :---: | :---: |
| Vanilla VAE | 28.41 M | 148.2 GB | 0.360 s | 800.0\* |
| VA-VAE | 28.41 M | 203.0 GB | 0.534 s | 2966\* |
| Our Stage 1 | 304.4 M | 157.5 GB | 0.402 s | 107.2 |
| Our Stage 2 | 304.4 M | 238.2 GB | 0.772 s | 377.4 |
| Our Stage 3 | 304.4 M | 159.0 GB | 0.412 s | 91.55 |

To compare encoder inference compute cost, we present a simplified version of Tab. 7 (first table below) and a simplified version of Tab. 8 (second table below) for convenience. We do not report results for Vanilla VAE because it has the same inference cost as VA-VAE. Please refer to the revised paper for the complete version that includes decoder cost. Our main observations are as follows:

1. **Peak GPU Memory (First Table).** At lower resolutions and smaller batch sizes, our encoder consumes more peak GPU memory than VA-VAE (ours: 1.372 GB, VA-VAE: 0.838 GB at 256 resolution with a batch size of 4). However, at higher resolutions and larger batch sizes, our encoder uses substantially less memory (ours: 3.015 GB, VA-VAE: 22.78 GB at 1024 resolution with a batch size of 8).
2. **Latency and GFLOPS (Second Table).** Overall, our tokenizer has higher encoding latency and GFLOPS than VA-VAE. The only exception is at 512 resolution, where our tokenizer achieves lower encoding latency.

####

**Table: Inference Memory Consumption Comparison.** We report the encoder peak GPU memory under different resolutions and batch sizes (batch size \= 1 / 4 / 8).

| Tokenizer | Input Resolution | Enc. \#Params | Enc. Peak GPU Memory (GB, bs=1 / 4 / 8\) |
| :---: | :---: | :---: | :---: |
| **VA-VAE** | 256 × 256 | 28.41 M | 0.301 / 0.838 / 1.543 |
| **Ours** | 256 × 256 | 304.4 M | 1.363 / 1.372 / 1.385 |
| **VA-VAE** | 512 × 512 | 28.41 M | 0.841 / 2.959 / 5.794 |
| **Ours** | 512 × 512 | 304.4 M | 1.284 / 1.450 / 1.674 |
| **VA-VAE** | 1024 × 1024 | 28.41 M | 2.958 / 11.45 / 22.78 |
| **Ours** | 1024 × 1024 | 304.4 M | 1.450 / 2.121 / 3.015 |

####

**Table: Inference Compute Cost Comparison.** We measure the GFLOPS and Latency under different image resolutions, using a single NVIDIA A100 GPU with a batch size of 1\.

| Tokenizer | Input Resolution | Enc. \#Params | Enc.  GFLOPS | Enc. Latency |
| :---: | :---: | :---: | :---: | :---: |
| VA-VAE | 256 × 256 | 28.41 M | 69.21 | 6.548 ms |
| Ours | 256 × 256 | 304.4 M | 77.85 | 13.42 ms |
| VA-VAE | 512 × 512 | 28.41 M | 279.2 | 19.44 ms |
| Ours | 512 × 512 | 304.4 M | 310.4 | 18.92 ms |
| VA-VAE | 1024 × 1024 | 28.41 M | 1155 | 73.92 ms |
| Ours | 1024 × 1024 | 304.4 M | 1241 | 121.2 ms |

\[1\] Yao, Jingfeng, et al. “Reconstruction vs. Generation: Taming Optimization Dilemma in Latent Diffusion Models.” CVPR, 2025\.

---

### Author Response · Authors · 2025-12-03
**Summary of the Rebuttal for AC**

We sincerely thank all the reviewers, area chairs, and program chairs for their time and thoughtful feedback. Here we provide **a concise summary for the area chairs**, highlighting our main contributions, the core reviewer concerns, and our corresponding updates and responses. Full, reviewer-specific replies are included in the main rebuttal.

Our key contribution is to **align a pretrained visual encoder**, rather than training a tokenizer from scratch, to serve as the tokenizer for latent diffusion models, producing semantically rich latents that significantly strengthen downstream diffusion training and generation quality. Reviewers highlighted the idea as “simple but effective” (XEQ8, GRL2, LVhM). They also emphasized the strength of our empirical validation, noting the usefulness of our ablations (XEQ8, GRL2), the value of our semantic and reconstruction tracking (XEQ8), and the stronger generation performance compared to VA-VAE \[1\] and FLUX-VAE \[2\] (GRL2). We are grateful that the overall experiments were viewed as comprehensive (LVhM).

The main or common concerns of reviewers include:

1. whether improved results come from a genuinely richer semantic space or merely from initialization of pretrained encoders (Reviewer XEQ8).
2. compute and memory cost comparison (Reviewers XEQ8, svcV, GRL2)
3. trade-off between semantics and reconstruction (Reviewers svcV, LVhM)
4. generalization beyond ImageNet for OOD testing (Reviewer LVhM)

We address the reviewers’ concerns in our responses to each reviewer. We have revised the paper and added new experiments and analyses. All newly added content is highlighted in blue for ease of review. The core updates are summarized in *[To All Reviewers](https://openreview.net/forum?id=ajnBafpqmE&noteId=rBMoKFIvCS)*.

\[1\] Yao, Jingfeng, et al. “Reconstruction vs. Generation: Taming Optimization Dilemma in Latent Diffusion Models.” CVPR, 2025\.
\[2\] Black Forest Labs. “FLUX.” GitHub Repository, 2024\. https://github.com/black-forest-labs/flux.

---

### Meta-Review · Area_Chair_gfZq · 2026-01-06

**Summary:**

This paper proposes a simple yet effective framework for constructing semantically richer image tokenizers by leveraging pretrained vision foundation models, particularly DINOv2, and introducing a carefully designed three-stage training pipeline. Reviewer XEQ8 highlights that the work addresses an important and timely problem: the semantic weakness of reconstruction-only VAE tokenizers, and commends the clarity of presentation, strong quantitative results, and especially the extensive ablations that track reconstruction and semantic capacity throughout training. Reviewer svcV notes that the overall framework is clear and empirically effective, reporting measurable improvements in linear probing accuracy and FID on ImageNet within a modern diffusion backbone. Reviewer GRL2 emphasizes that directly employing a pretrained vision foundation model followed by an adaptor is a novel and practically impactful design choice, yielding faster convergence and consistently better gFID across sampling steps and CFG scales compared to VA-VAE and FLUX VAE baselines. Reviewer LVhM further agrees that the approach is simple yet effective, with comprehensive experiments demonstrating the benefits of the three-stage design in balancing semantic preservation and reconstruction quality. Overall, despite some variance in perceived contribution level, the AC thinks the paper makes a solid contribution to generative models.

**Reviewer Concerns:**

While I don't see the feedback from the reviewers, to me, all concerns have been addressed well in the rebuttal.

**Reviewer Scores:**

-Reviewer XEQ8: Likely would have increased the score. The rebuttal clarifies the motivation behind the three-stage design, explains the placement of semantic regularization, and better frames the linear-probing drop as an explicit and controlled trade-off rather than a failure, reducing their main uncertainty.

- Reviewer svcV: Likely would have raised the score from 4 to around 6. The discussion helps address concerns about adaptor dimensionality sensitivity, pre-CFG performance, and the empirical reconstruction–generation trade-off, even though cost analysis and robustness questions remain.

- Reviewer GRL2: Likely would have increased the score from 4 to around 6. Additional clarification on convergence behavior, empirical advantages over VA-VAE and FLUX VAE, and the practical effectiveness of the pretrained-encoder-plus-adaptor design would alleviate some skepticism, though concerns about principled training and semantic analysis would persist.

- Reviewer LVhM: Likely to increase from 4 to 6). While the rebuttal may partially clarify motivation and empirical intent, major concerns about ImageNet-only evaluation, potential overfitting due to pretrained DINOv2, reconstruction degradation, and reproducibility are largely structural and unlikely to be fully resolved through discussion alone.

- Overall, the AC sees that the rebuttal is strong and could have resolved all remaining concerns.

---

### Decision · Program_Chairs · 2026-01-26

Accept (Poster)